# FOXP3 recognizes microsatellites and bridges DNA through multimerization

Wenxiang Zhang[1,2,4], Fangwei Leng[1,2,4], Xi Wang[1,2], Ricardo N. Ramirez[3], Jinseok Park[3], Christophe Benoist[3] & Sun Hur[1,2✉]

FOXP3 is a transcription factor that is essential for the development of regulatory T cells, a branch of T cells that suppress excessive inflammation and autoimmunity[1–5]. However, the molecular mechanisms of FOXP3 remain unclear. Here we here show that FOXP3 uses the forkhead domain—a DNA-binding domain that is commonly thought to function as a monomer or dimer—to form a higher-order multimer after binding to $T_nG$ repeat microsatellites. The cryo-electron microscopy structure of FOXP3 in a complex with $T_3G$ repeats reveals a ladder-like architecture, whereby two double-stranded DNA molecules form the two 'side rails' bridged by five pairs of FOXP3 molecules, with each pair forming a 'rung'. Each FOXP3 subunit occupies TGTTTGT within the repeats in a manner that is indistinguishable from that of FOXP3 bound to the forkhead consensus motif (TGTTTAC). Mutations in the intra-rung interface impair $T_nG$ repeat recognition, DNA bridging and the cellular functions of FOXP3, all without affecting binding to the forkhead consensus motif. FOXP3 can tolerate variable inter-rung spacings, explaining its broad specificity for $T_nG$-repeat-like sequences in vivo and in vitro. Both FOXP3 orthologues and paralogues show similar $T_nG$ repeat recognition and DNA bridging. These findings therefore reveal a mode of DNA recognition that involves transcription factor homomultimerization and DNA bridging, and further implicates microsatellites in transcriptional regulation and diseases.

How transcription factors (TFs) use a limited repertoire of DNA-binding domains (DBDs) to orchestrate complex gene regulatory networks is a central and yet unresolved question[6–9]. Although certain TFs, such as those with zinc-finger DBDs, can expand the complexity of their sequence specificity by forming an array of DBDs, the vast majority of TFs use a single DBD with narrow sequence specificity shared with other members of the DBD family[7]. One prominent model to rationalize this apparent paradox is that cooperative actions of multiple distinct TFs with distinct DBDs give rise to combinatorial complexity[10,11]. However, whether a single TF with a single DBD can also recognize distinct sequences on its own and perform divergent transcriptional functions, depending on the conformation or multimerization state, has not been fully addressed.

FOXP3 is an essential TF in regulatory T ($T_{reg}$) cell development, in which loss-of-function mutations cause a severe multiorgan autoimmune disease, immune dysregulation, polyendocrinopathy, enteropathy and X-linked (IPEX) syndrome[1–5]. Previous studies showed that FOXP3 remodels the global transcriptome and three-dimensional genome organization in the late stage of $T_{reg}$ cell development[12–15]. However, the molecular mechanisms of FOXP3, including its direct target genes and in vivo sequence specificity, remain unclear[13–16].

FOXP3 DNA binding is primarily mediated by a forkhead domain, which is shared among around 50 TFs of the forkhead family[17,18]. Most forkhead domains form a conserved winged-helix conformation

and recognize the forkhead consensus motif (FKHM) sequence (TGTTTAC)[19]. While the isolated forkhead domain of FOXP3 was originally crystallized as an unusual domain-swap dimer[20,21], a recent study showed that FOXP3 does not form a domain-swap dimer but, instead, folds into the canonical winged-helix conformation in the presence of the adjacent RUNX1-binding region (RBR)[22]. It was further shown that FOXP3 has a strong preference for inverted-repeat FKHM (IR-FKHM) over a single FKHM in vitro by forming a head-to-head dimer[22]. However, previous chromatin immunoprecipitation followed by sequencing (ChIP–seq)[14,23,24] and cleavage under targets and release using nuclease sequencing (CNR-seq)[14] analyses did not reveal enrichment of IR-FKHM in FOXP3-occupied genomic regions within cells[22]. While individual FKHM is present in around 10% of the FOXP3 ChIP peaks, they too may not be the FOXP3-binding sites, as DNase I protection patterns at these sites were unaffected by *FOXP3* deletion[24]. These observations raised the question of what sequences FOXP3 in fact recognizes in cells and whether FOXP3 can use a previously unknown mode of binding to recognize new sequence motifs that are distinct from FKHM.

## FOXP3 binds to $T_nG$ repeat microsatellites

To re-evaluate FOXP3 sequence specificity, we performed an unbiased pull-down of genomic DNA with recombinant FOXP3 protein. The use of genomic DNA, as opposed to synthetic DNA oligos, enables the testing

[1]Howard Hughes Medical Institute and Program in Cellular and Molecular Medicine, Boston Children's Hospital, Boston, MA, USA. [2]Department of Biological Chemistry and Molecular Pharmacology, Blavatnik Institute, Harvard Medical School, Boston, MA, USA. [3]Department of Immunology, Blavatnik Institute, Harvard Medical School, Boston, MA, USA. [4]These authors contributed equally: Wenxiang Zhang, Fangwei Leng. ✉e-mail: Sun.Hur@crystal.harvard.edu

of sequence specificity in the context of a naturally existing repertoire of sequences. It can also enable identification of longer motifs by using genomic DNA fragments longer than around 20–40 bp—the typical lengths used in previous biochemical studies of FOXP3[22,25,26]. We isolated genomic DNA from mouse EL4 cells, fragmented to about 100–200 bp, incubated with purified, MBP-tagged mouse FOXP3 and performed MBP pull-down, followed by next-generation sequencing (NGS) of the co-purified DNA (FOXP3 PD-seq; Fig. 1a). We used recombinant FOXP3 protein (FOXP3($\Delta$N)) containing the zinc-finger, coiled-coil, RBR and forkhead domains but lacking the N-terminal proline-rich region (Fig. 1a). FOXP3($\Delta$N) was shown to display the same DNA specificity as full-length FOXP3 among the test set[22]. De novo motif analysis showed a strong enrichment of $T_nG$ repeats ($n$ = 2–5) by FOXP3 pull-down, using either pull-down of MBP alone or the input as a control (Fig. 1b and Supplementary Table 1a). The $T_3G$ repeat sequence was the highest-ranking motif, accounting for 49.8% of the peaks. No other motifs, including the canonical FKHM or other repeats, were similarly enriched (Supplementary Table 1a). FOXP3 pull-down using nucleosomal DNA from mouse EL4 cells revealed similar enrichment of $T_nG$-repeat-like sequences (Supplementary Table 1a).

De novo motif analysis of previously published FOXP3 CNR-seq[12,14] and ChIP–seq data[14,23] also identified $T_nG$-repeat-like motifs as one of the most significant motifs in all four datasets (Fig. 1b and Supplementary Table 1b). The enrichment score for $T_nG$-repeat-like motifs ($E$ value) was more significant than that of FKHM in all cases (Fig. 1b and Supplementary Table 1b). Note that $T_nG$-repeat-like motifs have not been reported from these original studies, probably reflecting the common practice of discarding simple repeats in motif analysis. $T_nG$-repeat-like motifs were not identified from open chromatin regions (as measured using the assay for transposase-accessible chromatin with sequencing (ATAC–seq))[27] in $T_{reg}$ cells that were not occupied by FOXP3 (Supplementary Table 1b).

To examine whether $T_nG$-repeat-like sequences indeed contribute to FOXP3–DNA interaction in $T_{reg}$ cells, we analysed published FOXP3 CNR-seq data generated using $F_1$ hybrids of the C57BL/6J (B6) and CAST/ EiJ (Cast) mouse strains[14]. Owing to the wide divergence between the B6 and Cast mouse genomes, such data enable the evaluation of the impact of sequence variations on TF binding. Out of 196 sites showing allelic imbalance (fold change ≥ 4) in FOXP3 CNR-seq, 76 sites contained $T_nG$-repeat-like elements in at least one allele, the frequency (38.8%) significantly higher than that in the mouse genome (around 0.06%, $P < 1 \times 10^{-8}$; Extended Data Fig. 1a). Furthermore, all but four sites showed a $T_nG$ repeat length mirroring the allelic bias in FOXP3 occupancy (Fig. 1c,d). Of the 76 sites, we randomly chose 50 sites, 25 each from B6- and Cast-biased peaks, and tested the FOXP3-binding efficiency using a FOXP3($\Delta$N) pull-down assay. Out of the 50 pairs of sequences tested, the pull-down efficiency of 47 pairs recapitulated differential binding in CNR-seq (Fig. 1c,e). All 47 sites showed significant truncations in the $T_nG$ repeats in the less-preferred allele (the full list of sequences is provided in Supplementary Table 2a). Note that the pull-down preference for longer $T_nG$ repeats was not due to the different DNA lengths used—an extension of the less-preferred allele sequences with a random sequence at a DNA end (Fig. 1c (B6* and Cast*); the sequence is provided in Supplementary Table 2b) did not alter the allele bias. Together, these results suggest that $T_nG$-repeat-like elements have an important role in FOXP3–DNA interaction in vitro and in vivo.

Genome-wide analysis showed that there are 46,574 loci in the *Mus musculus* genome with $T_nG$-repeat-like sequences and that they are predominantly located distal to annotated transcription start sites (TSSs), with 9.5% residing within 3 kb of the annotated TSSs (Extended Data Fig. 1a,b). By contrast, among the $T_nG$-repeat-containing FOXP3 CNR peaks[12,14] ($n$ = 3,301 out of the 9,062 CNR peaks), 38.4% were found within 3 kb of TSSs (Extended Data Fig. 1c). $T_nG$-repeat-containing FOXP3 CNR peaks also displayed higher levels of trimethylated H3K4 (H3K4me3), acetylated H3K27 (H3K27ac) and chromatin accessibility

compared with the genome-wide $T_nG$ repeats (Extended Data Fig. 1d–f). These results suggest that, although $T_nG$-repeat-like sequences are common in the *M. musculus* genome, FOXP3 uses a small fraction of $T_nG$-repeat-like sequences in accessible, functional sites for transcriptional regulation.

## FOXP3 multimerizes on $T_nG$ repeats

To examine whether the $T_nG$ repeat enrichment in PD-seq and CNR/ ChIP–seq represents previously unrecognized sequence specificity of FOXP3, we compared FOXP3 binding to DNA with $T_nG$ repeats ($n$ = 1–6) versus those containing IR-FKHM, the highest-affinity sequence reported for FOXP3 to date[22]. All DNAs were 45 bp long (the sequences are provided in Supplementary Table 2b). We found that the $T_3G$ repeat was comparable to IR-FKHM in FOXP3 binding and was the tightest binder among the $T_nG$ repeats (Fig. 1f), consistent with it being the most significant motif in PD-seq (Fig. 1b). The $T_2G$, $T_4G$ and $T_5G$ repeats also showed more efficient binding than a single FKHM (1×FKHM) or random sequence (no FKHM). No other simple repeats showed FOXP3 binding comparable to $T_3G$ repeats (Fig. 1g). FOXP3 affinity increased with the copy number of $T_3G$ when compared among DNAs of the same length (Fig. 1h). The preference for $T_3G$ repeats was also observed using full-length FOXP3 expressed in HEK293T cells (Extended Data Fig. 1g) or when the pull-down bait was switched from FOXP3 to DNA (Extended Data Fig. 1h). Finally, FOXP3 can bind to $T_3G$ repeats even in the presence of nucleosomes (Extended Data Fig. 1i), suggesting that similar interactions can occur in the context of chromatinized DNA.

We next investigated how FOXP3 recognizes $T_3G$ repeats. In contrast to IR-FKHM, $T_3G$ repeat DNA induced FOXP3 multimerization as indicated by slowly migrating species in native gel-shift assay (Fig. 1i). Protein–protein cross-linking also suggested higher-order multimerization in the presence of $T_3G$ repeats, but not with IR-FKHM or 1×FKHM (Extended Data Fig. 1j). In support of $T_3G$-repeat-induced multimerization, MBP-tagged FOXP3 co-purified with GST-tagged FOXP3 only in the presence of $T_3G$ repeats, but not with IR-FKHM (Extended Data Fig. 1k). Finally, negative electron microscopy revealed a filamentous multimeric architecture of FOXP3 on 36 tandem repeats of $T_3G$ (Fig. 1j), the copy number chosen to aid clear visualization. Other DNAs of the same length, such as $(A_3G)_{36}$, $(TGTG)_{36}$ or $(IR-FKHM)_5$, did not show similar multimeric architectures under the equivalent conditions (Fig. 1j and Extended Data Fig. 1l). These results suggest that FOXP3 forms distinct multimers on $T_3G$ repeats.

## The structure of FOXP3 bound to $T_3G$ repeats

To understand how FOXP3 forms multimers on $T_3G$ repeats, we determined the cryo-electron microscopy (cryo-EM) structure of FOXP3($\Delta$N) in a complex with $(T_3G)_{18}$. Single-particle reconstruction led to a 3.6-Å-resolution map after global refinement and a 3.3-Å-resolution map after focused refinement of the central region (Extended Data Fig. 2a–f and Extended Data Table 1). The density map revealed two continuous double-stranded DNA molecules spanning about 50 bp (Fig. 2a). Both DNA molecules adopted the classic B-form DNA with an average twist angle of 33.5° per bp and an average rise of 3.19 Å per bp. The density map could also be fitted with the crystal structure of DNA-bound FOXP3 monomer containing part of RBR and forkhead (residues 326–412), enabling placement of ten FOXP3 subunits without zinc-finger, coiled-coil and RBR residues 188–325. Only the non-swap, winged-helix conformation was compatible with the density map (Extended Data Fig. 2g). Consistent with this, FOXP3($\Delta$N/R337Q), a loss-of-function IPEX mutation that induces domain-swap dimerization[22], showed significantly reduced affinity for $T_3G$ repeats (Extended Data Fig. 2h).

DNA sequence assignment (Extended Data Fig. 3a) revealed that all ten FOXP3 subunits interacted with the $T_3G$ repeat DNA in a manner that was indistinguishable from that of FOXP3 bound to the canonical FKHM,

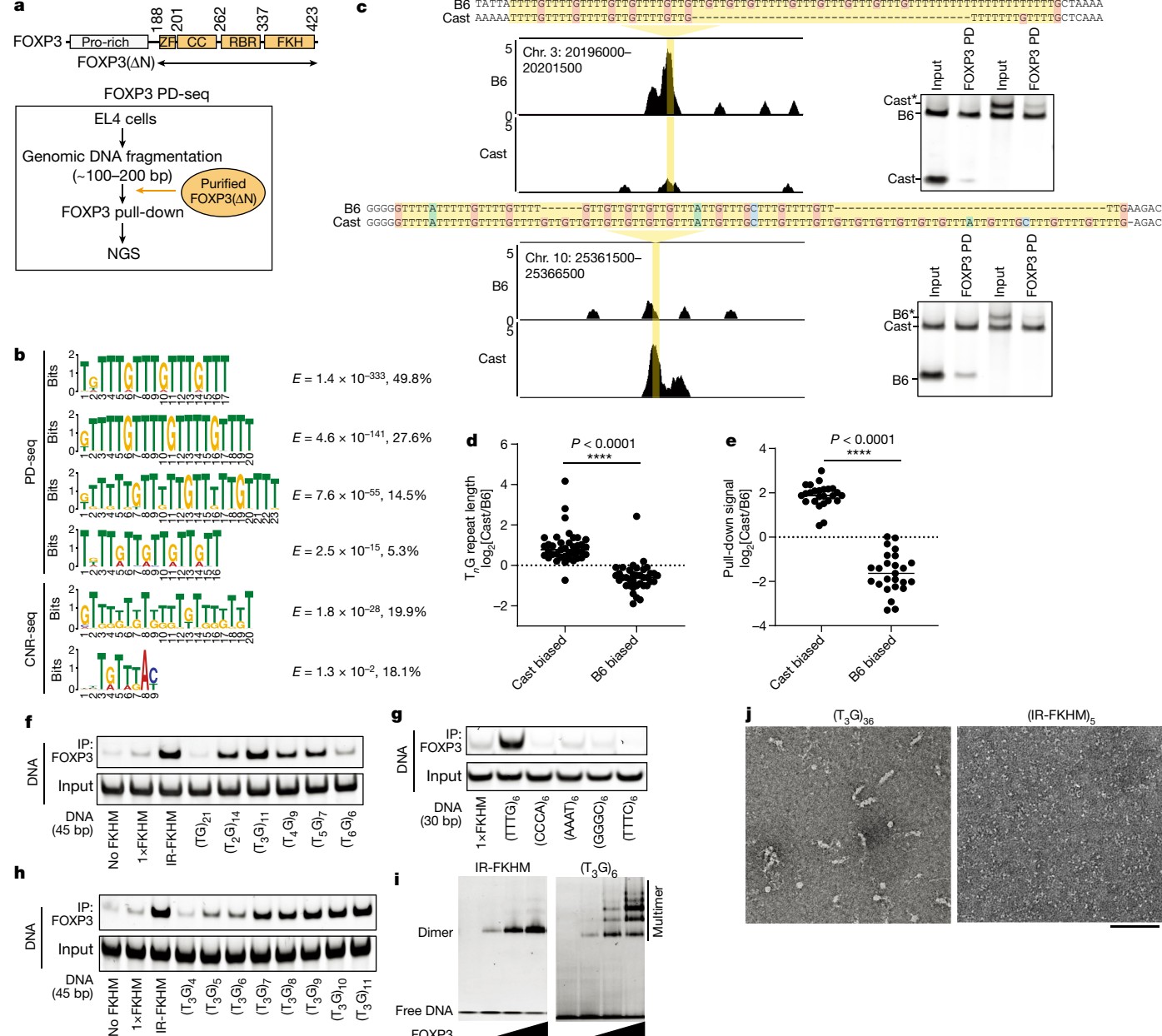

**Fig. 1 | FOXP3 recognizes $T_nG$ repeat microsatellites. a**, The FOXP3 domain architecture and schematic of FOXP3 PD-seq. CC, coiled-coil domain; ZF, zinc finger domain. **b**, De novo motif analysis of FOXP3 PD-seq peaks ($n = 21,605$) and CNR-seq peaks[14] ($n = 6,655$) using MEME-ChIP and STREME. The *E* score and the percentage of peaks containing the given motif are shown on the right. See Supplementary Table 1a,b for the comprehensive list of motifs for PD-seq, CNR-seq[12,14] and ChIP–seq data[14,23]. **c**, Allelic imbalance in FOXP3 binding in vivo. Left, genome browser view of CNR-seq[14], showing B6-biased (top) and Cast-biased (bottom) peaks. B6 genomic coordinates are shown at the top left. Chr., chromosome. **d**, $T_nG$ repeat length comparison between Cast and B6 mice at 76 loci, showing allelic bias in **c**. Repeat lengths were measured in nucleotides. $n = 39$ (Cast-biased loci) and $n = 37$ (B6-biased loci) were used for this comparison. Statistical analysis was performed using two-tailed unpaired

*t*-tests; ****$P < 0.0001$. **e**, Allelic imbalance in FOXP3 binding in vitro. A total of 50 pairs of Cast and B6 sequences (Supplementary Table 2a) was chosen from the 76 pairs in **d** and analysed using FOXP3(ΔN) pull-down. For each pair, the recovery rate of the Cast and B6 DNA was measured and their ratios were plotted. Each datapoint represents the average of the two pull-downs. Statistical analysis was performed using two-tailed unpaired *t*-tests. **f**, FOXP3–DNA interaction was measured using FOXP3(ΔN) pull-down. DNA containing a random sequence (no FKHM), a single FKHM (1×FKHM), IR-FKHM or tandem repeats of $T_nG$ ($n = 1$–$6$) were used. All DNAs were 45 bp long. **g**, FOXP3–DNA interaction using DNAs (30 bp) containing various tandem repeats, including T3G repeats. **h**, FOXP3–DNA interaction using DNAs (45 bp) containing 4–11 repeats of T3G. **i**, Native gel shift assay of MBP-tagged FOXP3(ΔN) (0–0.4 μM) with DNA (30 bp, 0.05 μM) containing IR-FKHM or $(T_3G)_6$. **j**, Representative negative-stain EM images of FOXP3(ΔN) in a complex with $(T_3G)_{36}$ and (IR-FKHM)$_5$. Both DNAs were 144 bp long. Scale bar, 100 nm.

recognizing TGTTTGT in place of TGTTTAC (Fig. 2b). This FOXP3–DNA register was further confirmed by FOXP3 footprint analysis using DNA mutagenesis and NFAT–FOXP3 cooperativity (Extended Data Fig. 3b,c).

Note that NFAT is a known interaction partner of FOXP3 and assists FOXP3 binding to DNA only when their binding sites are 3 bp apart, the property used for inferring FOXP3–DNA registers (Extended Data Fig. 3b).

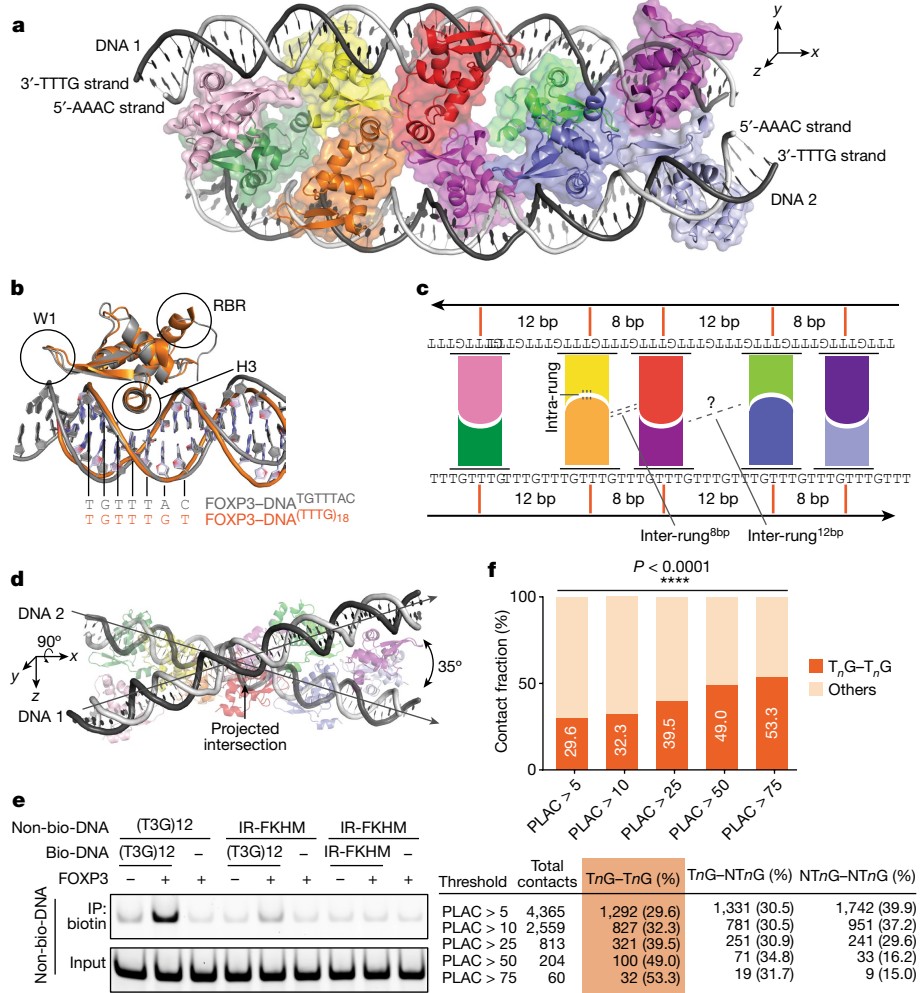

**Fig. 2 | FOXP3 forms a ladder-like multimer after binding to T₃G repeat DNA.**
**a**, The cryo-EM structure of a FOXP3(ΔN) decamer in a complex with two DNA molecules (grey) containing (T₃G)₁₈. Each of the ten FOXP3 subunits are coloured differently. **b**, Comparison of a representative FOXP3(ΔN) subunit from **a** (orange) with a FOXP3(ΔN) subunit from the head-to-head dimeric structure (grey; Protein Data Bank (PDB): 7TDX). H3 recognizes the DNA sequence (TGTTTAC in the head-to-head dimer, TGTTTGT in the ladder-like multimer) by inserting it into the major groove. **c**, Schematic of the ladder-like architecture of FOXP3 on T₃G-repeat DNA. **d**, The skew relationship between the two DNA molecules, which is evident when looking down the y axis of **a**. **e**, DNA-bridging assay. Biotinylated DNA (bio-DNA, 82 bp) and non-biotinylated DNA

(non-bio-DNA, 60 bp) were mixed at a 1:1 ratio (0.1 μM each), incubated with FOXP3(ΔN) (0.4 μM) and processed for Streptavidin pull-down before gel analysis. Non-biotinylated DNA in the eluate was visualized by SybrGold staining. **f**, Chromatin contacts at FOXP3-bound anchors identified using Hi-C-seq and PLAC-seq[12]. Contacts with a frequency of >5 in the WT T_reg cell Hi-C analysis and connected by two FOXP3-bound anchors were analysed with an increasing FOXP3 PLAC-seq count threshold. The percentage of the unique contacts mediated by two T_nG anchors (out of all unique contacts between two FOXP3-bound anchors) is indicated. All T_nG–T_nG contacts were between two distinct 10 kb anchor bins. NT_nG, non-T_nG.

The overall architecture resembled a ladder whereby the two double-stranded DNA molecules formed side rails bridged by five rungs, each of which consisted of two FOXP3 subunits bound to different DNA and joined by direct protein–protein interactions (intra-rung interactions) (Fig. 2a,c). These rungs were separated by 8 bp or 12 bp in an alternating manner, forming two different types of inter-rung interactions (inter-rung^8bp and inter-rung^12bp) with divergent significance (discussed below). Given that both DNA molecules had the helical periodicity of 10.7 bp per turn, this alternating spacing pattern enabled FOXP3 molecules to occupy consecutive major grooves on one side of each DNA. This geometry, in turn, enabled the FOXP3 molecules on opposing DNA to face each other and form the rungs of the ladder. None of the intra- and inter-rung interactions resembled the previously reported head-to-head dimerization interaction[22] (Extended Data Fig. 3d), revealing a distinct mode of molecular assembly for FOXP3.

The two DNA molecules are skew to each other (that is, non-parallel, non-intersecting). When projected onto the xy plane as in Fig. 2a, they appeared parallel, but projection onto the xz plane as in Fig. 2d suggested that they approached each other at an angle of 35°. The divergence of the two DNA molecules can explain why the multimeric assembly was limited to the decamer spanning around 50 bp near the projected intersection point (Fig. 2d), even though the DNA sample in cryo-EM was 72 bp long and had many more T₃G repeats to accommodate additional FOXP3 molecules. The lack of cryo-EM density for FOXP3 molecules bound to DNA without forming the rung suggests that the intra-rung interaction is critical for stable protein–DNA interactions. In other words, DNA bridging may be an integral part of the assembly.

To test whether DNA bridging indeed occurs in solution, we examined co-purification of non-biotinylated DNA (prey) with biotinylated DNA (bait) in the presence and absence of FOXP3. We observed DNA bridging

between biotinylated and non-biotinylated $T_3G$ repeats only in the presence of FOXP3($\Delta$N) (Fig. 2e). DNA bridging was not observed between IR-FKHM and IR-FKHM DNAs or between $(T_3G)_{12}$ and IR-FKHM DNAs. Similar $T_nG$-repeat-dependent bridging was observed with full-length FOXP3 expressed in HEK293T cells (Extended Data Fig. 3e). Moreover, $T_3G$-repeat DNA bridging occurred more efficiently with an increasing concentration of FOXP3 (Extended Data Fig. 3f), suggesting that DNA bridging is not an artificial consequence of saturating multimeric FOXP3 with DNA.

To further examine whether FOXP3 binding to $T_nG$ repeats mediates long-distance chromatin contacts in $T_{reg}$ cells, we analysed the available Hi-C-seq, PLAC-seq and Hi-C coupled with ChIP–seq (HiChIP–seq) data[12,13]. The limited resolution of these data (5–10 kb) precluded direct motif analysis of the chromatin contact anchors. Instead, we examined how frequently contacts are made between anchors containing FOXP3 CNR peaks with $T_nG$ repeats ($T_nG$ anchors) versus those lacking $T_nG$ repeats (non-$T_nG$ anchors). Among the high-frequency contacts (Hi-C frequency > 5, PLAC frequency > 5–75) between FOXP3-bound anchors, we found that those between two $T_nG$ anchors (30–53%) were more prevalent than expected by chance (13.7%) and that such $T_nG$–$T_nG$ contacts were more enriched among the stronger contacts (Fig. 2f and Supplementary Table 3 (tabs 1–6)). By contrast, non-$T_nG$–non-$T_nG$ contacts were more depleted among the stronger contacts. This is despite the fact that non-$T_nG$ CNR peaks have higher levels of chromatin accessibility and H3K4me3 than $T_nG$ CNR peaks, while displaying similar H3K27ac levels (Extended Data Fig. 4a–c). Most of the $T_nG$–$T_nG$ contacts showed increased frequency in WT $T_{reg}$ cells relative to in *FOXP3*-knockout $T_{reg}$-like cells from mice (Extended Data Fig. 4d). Furthermore, many of the anchors for the $T_nG$–$T_nG$ contacts were near $T_{reg}$ cell signature genes (such as *Il2ra*, *Cd28*, *Tnfaip3* and *Ets1*; Supplementary Table 3 (tab 7)), and overlapped with previously characterized enhancer–promoter loop anchors in $T_{reg}$ cells (Extended Data Fig. 4e), implicating their transcriptional functions. These results together support that FOXP3 multimerization on $T_nG$ repeats contributes to long-distance chromatin contacts in $T_{reg}$ cells.

## Intra-rung interaction is essential

Examination of the intra-rung interactions showed that multiple distinct parts of the protein are involved; wing 1 (W1), a loop between helix 2 and 4 (H2/H4 loop) and helix 6 (H6) of one subunit interacted with RBR and H2/H4 loop of the other subunit within the rung (Fig. 3a). While the resolution at the interface was insufficient to assign precise side-chain conformations, the structure identified Arg356 in the H2/H4 loop; Val396 and Val398 in W1; and Asp409, Glu410 and Phe411 in H6 as residues at the interface (Fig. 3a). We also chose Val408 in H6, which was adjacent to the interface residues and is mutated to Met in a subset of patients with IPEX[15,28,29]. Mutations of these interface residues, including V408M, disrupted $T_3G$-repeat binding (Fig. 3b (right)) and DNA bridging (Fig. 3c). The same mutations had a minimal impact on IR-FKHM binding (Fig. 3b (left)), which requires head-to-head dimerization of FOXP3[22]. This is consistent with the previous structure showing that these residues are far from either the DNA binding or the head-to-head dimerization interface[22]. The negative effect of the intra-rung mutations on $T_3G$ repeat binding as well as DNA bridging further supports that DNA bridging is required for FOXP3 multimerization on $T_3G$ repeats, rather than a simple consequence of FOXP3 multimerization.

These intra-rung mutations disrupted cellular functions of FOXP3, as measured by FOXP3-induced gene expression (for example, CTLA4 and CD25 protein levels (Fig. 3d) and genome-wide mRNA levels (as measured by FOXP3 mRNA-seq in Extended Data Fig. 5a)), target loci binding (as measured by FOXP3 ChIP–seq in Extended Data Fig. 5b) and T-cell-suppressive functions (Fig. 3e). None of these mutations affected nuclear localization, the level of FOXP3 (Extended Data Fig. 5c,d) or

FOXP3's interaction with NFAT (Extended Data Fig. 5e), although a slight reduction in NFAT binding was seen for V398E. These results suggest that the ladder-like assembly is important for the transcriptional functions of FOXP3.

## Relaxed sequence specificity of multimer

We next examined the potential role of the inter-rung interactions. The inter-rung[8bp] interaction was mediated by RBR–RBR contacts, which displayed continuous EM density indicative of a strong ordered interaction (Fig. 4a and Extended Data Fig. 2f). In contrast to the intra-rung interface mutations, mutations in RBR, for example F331D, disrupted FOXP3 binding to both $T_3G$ repeats and IR-FKHM[22] (Fig. 4b and Extended Data Fig. 6a), suggesting that the RBR has an important role in both ladder-like multimerization and head-to-head dimerization[22]. Consistent with the importance of the inter-rung[8bp] interaction, changes in the inter-rung[8bp] spacing from 8 bp (1 bp gap) to 9 bp (2 bp gap) or 7 bp (no gap) resulted in a significant impairment in FOXP3 binding to $T_3G$ repeats (Fig. 4c).

In contrast to the inter-rung[8bp] interaction, the cryo-EM density for the inter-rung[12bp] interaction was difficult to interpret, which could reflect a weak or less-ordered interaction. In keeping with this, FOXP3 binding tolerated a wide range of inter-rung[12bp] spacings, with equivalent affinity observed for spacings of 11–13 bp (Fig. 4d). Notably, although 14–19 bp spacings were not tolerated, DNA with 21–22 bp spacings showed moderate binding. Given that 11–13 bp, 14–19 bp and 21–22 bp spacings would place FOXP3 one, one and a half and two helical turns away from the upstream FOXP3 molecule, respectively, this cyclical pattern suggests that the precise positions of FOXP3 are not essential for multimeric assembly, so far as the DNA sequence allows FOXP3 molecules to line up on one side of DNA and form the rungs. Consistent with this idea, DNA-bridging activity showed a similar cyclical pattern (Extended Data Fig. 6b).

This architectural flexibility may explain our observations in Fig. 1, which showed that FOXP3 could bind to a broad range of $T_nG$-repeat-like sequences besides perfect $T_3G$ repeats. These include tandem repeats of $T_2G$, $T_4G$ and $T_5G$ and their various mixtures found in the CNR-seq peaks with allelic imbalance (Supplementary Table 2a). To examine whether a similar ladder-like architecture forms with $T_nG$-repeat-like sequences that are not perfect $T_3G$ repeats, we used DNA-bridging activity as a measure of the ladder-like assembly. All 47 pairs of the DNA sequences showing allelic bias in FOXP3 binding in vivo and in vitro displayed the same allelic bias in DNA bridging (Fig. 4e and Supplementary Table 2a). The multimerization-specific IPEX mutation V408M abrogated bridging of $T_2G$, $T_4G$ and $T_5G$ repeat DNAs (Fig. 4f), suggesting a similar multimeric architecture for FOXP3 regardless of the exact $T_nG$ repeat sequences. Notably, suboptimal $T_nG$ repeats ($n = 2, 4, 5$) were bridged with $T_3G$ repeats more efficiently than with themselves (Fig. 4g and Extended Data Fig. 6c), suggesting that having a strong DNA as a bridging partner helps FOXP3 binding to suboptimal sequences. These results reveal yet another layer of complexity that can broaden the sequence specificity of FOXP3.

## $T_nG$ repeat binding is conserved in FOXPs

The studies above were performed using FOXP3 and $T_nG$-repeat-like elements from *M. musculus*. We next examined whether $T_nG$ repeat recognition by FOXP3 is conserved in other species besides *M. musculus*. Inspection of $T_nG$-repeat-like elements in the *Homo sapiens* and *Danio rerio* genomes revealed 18,164 and 5,517 distinct sites containing $T_nG$ repeats (>29 nucleotides), respectively, in comparison to the 46,574 sites in the *M. musculus* genome (Extended Data Fig. 1a). While $T_nG$-like repeats are more frequently located distal to TSSs in all three genomes of *H. sapiens*, *M. musculus* and *D. rerio*, greater fractions are located within around 3 kb of TSSs in higher eukaryotes (12.66%, 9.50% and

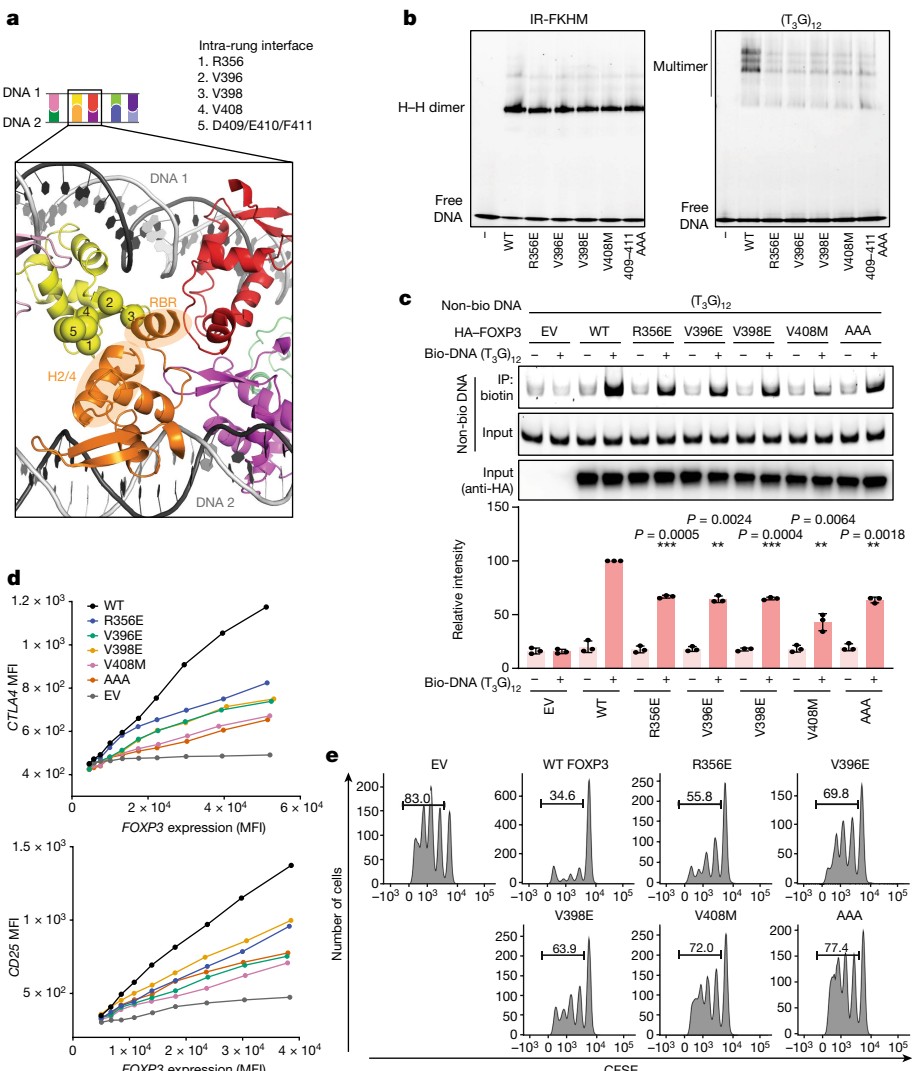

**Fig. 3 | Intra-rung interactions are essential for $T_nG$ repeat recognition, DNA bridging and the cellular functions of FOXP3. a**, The intra-rung interface. The α-carbons of Arg356, Val396, Val398, Val408 and Asp409/Glu410/Phe411 are shown as spheres. These residues on the yellow subunit interact with RBR and H2/H4 loop of the orange subunit. The subunit colours are as described in Fig. 2a. **b**, The effect of intra-rung interface mutations on DNA binding. MBP-tagged FOXP3(ΔN) (0.4 μM) was incubated with IR-FKHM or $(T_3G)_{12}$ (60 bp for both) and analysed using a native gel shift assay. **c**, The effect of intra-rung interface mutations on DNA bridging. FOXP3 (or empty vector (EV)) was expressed in HEK293T cells and the lysate was incubated with a mixture of biotinylated and non-biotinylated DNA (1:1 ratio) and then analysed using Streptavidin pull-down and gel analysis. The relative levels of non-biotinylated DNA co-purified with biotinylated DNA were quantified from three independent pull-downs. The difference was compared with the WT in the presence of biotinylated DNA. Statistical analysis was performed using two-tailed paired *t*-tests; ****P* < 0.001, \*\**P* < 0.005. **d**, Transcriptional activity of FOXP3. CD4[+] T cells were retrovirally transduced to express FOXP3, and its transcriptional activity was analysed by measuring the protein levels of the known target genes *CTLA4* and *CD25* using fluorescence-activated cell sorting (FACS). *FOXP3* levels were measured on the basis of Thy1.1 expression, which is under the control of IRES, encoded by the bicistronic *FOXP3* mRNA. MFI, mean fluorescence intensity. **e**, T cell suppression assay of intra-rung interface mutations. FOXP3-transduced T cells (suppressors) were mixed with naive T cells (responders) at a 1:2 ratio and the effect of the suppressor cells on the proliferation of the responder cells was measured on the basis of the carboxyfluorescein succinimidyl ester (CFSE) dilution profile of the responder T cells.

5.72% for *H. sapiens*, *M. musculus* and *D. rerio*, respectively) (Extended Data Fig. 1b), even though all three species have similar gene-to-genome size ratios (Extended Data Fig. 1a (top)). This observation suggests that $T_nG$ repeats may have been coopted for transcriptional functions in higher eukaryotes.

We examined FOXP3 from *H. sapiens*, *Ornithorhynchus anatinus* and *D. rerio*. All three FOXP3 orthologues showed preferential binding to $T_3G$ repeats and IR-FKHM in comparison to a single FKHM or no FKHM (Extended Data Fig. 6d). They also bridged $T_3G$ repeats (Extended Data Fig. 6e), suggesting a ladder-like assembly similar to that of *M. musculus* FOXP3. This is in keeping with the fact that the key residues for multimerization were broadly conserved or interchanged with similar amino acids in FOXP3 orthologues (Extended Data Fig. 6f). Given that *D. rerio* FOXP3 represents one of the most distant orthologues from mammalian FOXP3, these results suggest that $T_nG$ repeat recognition and ladder-like assembly may be ancient properties of FOXP3.

Inspection of the sequence alignment of forkhead TFs revealed that the key residues for multimerization are also well conserved within the FOXP family, but not outside (Fig. 5a). Biochemical analysis of *M. musculus* FOXP1, FOXP2 and FOXP4 in the FOXP family showed that they preferentially bound to $T_3G$ repeats and bridged $T_3G$ repeat DNA as with FOXP3 (Fig. 5b,c and Extended Data Fig. 6g). De novo motif analysis of previously published ChIP–seq data showed that $T_nG$-repeat-like

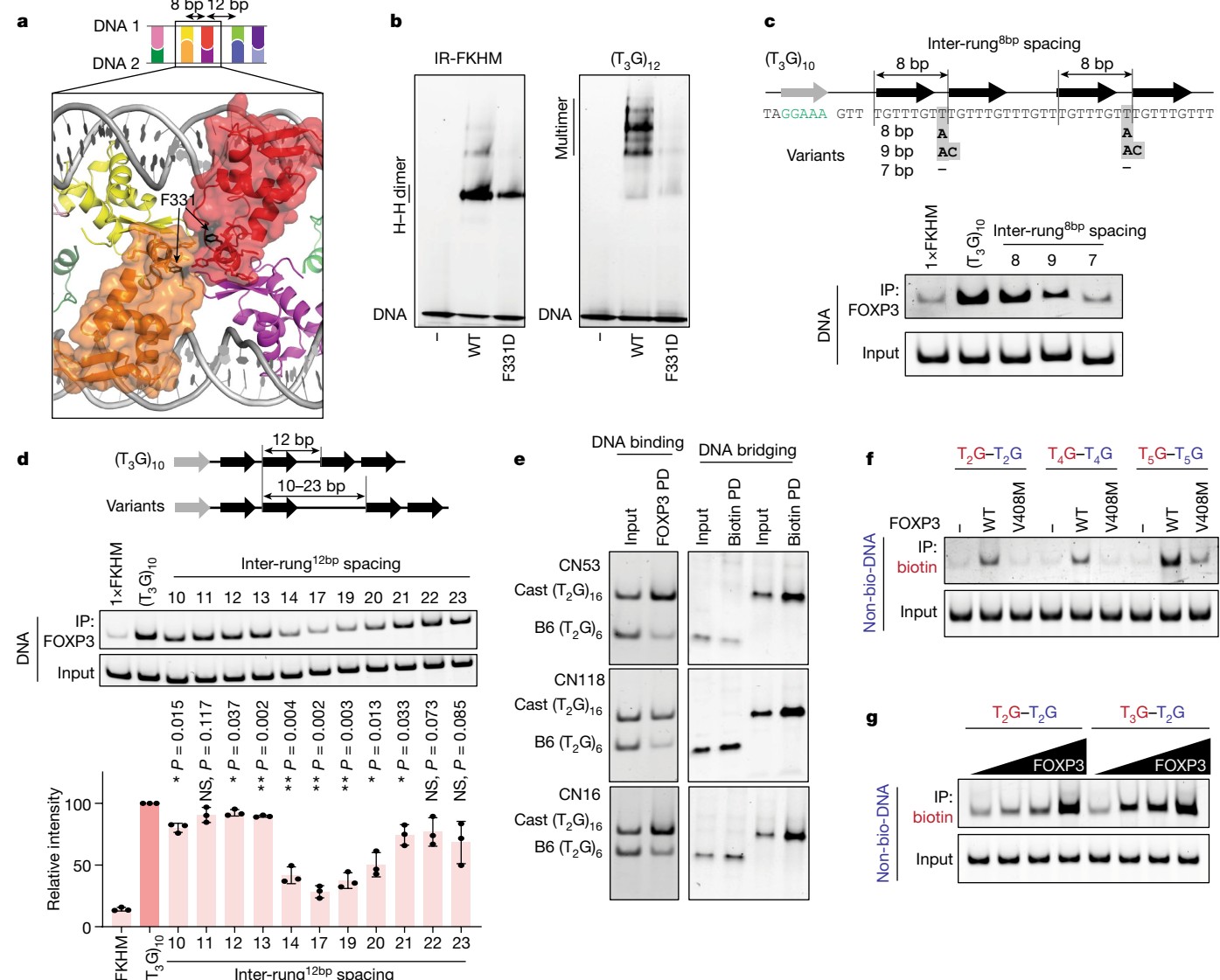

**Fig. 4 | Architectural flexibility of the ladder-like assembly broadens the sequence specificity of FOXP3. a**, Structure highlighting the inter-rung[8bp] interactions between the orange and red subunits (in surface representation). The interactions are primarily between RBRs, where Phe331 resides. **b**, The effect of the inter-rung[8bp] mutation (F331D) on the FOXP3–DNA interaction was analysed using FOXP3(ΔN) pull-down. **c**, The effect of inter-rung[8bp] spacings on FOXP3 binding was determined using FOXP3(ΔN) pull-down. Both inter-rung[8bp] gap nucleotides were changed from T in $(T_3G)_{10}$ to A (8 bp spacing), to AC (9 bp spacing) or to no nucleotide (7 bp spacing). The black arrows indicate FOXP3 footprints. The grey arrow and green-coloured nucleotides indicate the NFAT-binding site. NFAT interacts with FOXP3 and helps in fixing the FOXP3–DNA register, which was necessary to examine the effect of DNA sequence variations at or between the FOXP3 footprints. **d**, The effect of inter-rung[12bp] spacings on FOXP3 binding was analysed using FOXP3(ΔN)

pull-down. The inter-rung[12bp] spacing was changed from 12 bp in $(T_3G)_{10}$ to 10–23 bp (the sequences are provided in Supplementary Table 2b). The average recovery rate of DNA from three independent pull-downs was plotted. Statistical analysis was performed using two-tailed paired $t$-tests in comparison to $(T_3G)_{10}$; *$P < 0.05$; NS, $P > 0.05$. **e**, Comparison of Cast and B6 sequences in FOXP3 binding (left) and DNA bridging (right). Three pairs of sequences at the loci CN53, CN118 and CN16 with Cast bias in the CNR-seq analysis were compared (Supplementary Table 2a) using FOXP3(ΔN) pull-down. **f**, DNA bridging between $(T_2G)_{14}$ and $(T_2G)_{14}$, between $(T_4G)_9$ and $(T_4G)_9$, and between $(T_5G)_7$ and $(T_5G)_7$ in the presence of WT FOXP3 or the IPEX mutant V408M. Biotinylated and non-biotinylated DNA are coloured red and blue, respectively. **g**, DNA bridging between $(T_2G)_{14}$ and $(T_2G)_{14}$, and between $(T_2G)_{14}$ and $(T_3G)_{11}$ by FOXP3 (0–0.4 μM).

motifs were indeed enriched in FOXP1- and FOXP4-occupied sites (Fig. 5d; the full list and references are provided in Supplementary Table 1c). This feature was particularly strong for FOXP1 in lymphoma cell lines (SU-DHL-6 and U-2932) and mouse neural stem cells—the $T_nG$-repeat-like motif was the most significant motif, whereas FKHM ranked far lower (Fig. 5d). However, in the VCap and K-562 cell lines, FOXP1 ChIP–seq peaks did not show $T_nG$-like elements, although FKHM was identified as one of the most significant motifs in these cells

(Supplementary Table 1c). Similar context-dependent enrichment of $T_nG$-repeat-like elements was seen with FOXP4, although the motif enrichment was not as strong as with FOXP1 or FOXP3 (Fig. 5d and Supplementary Table 1c). By contrast, long (>10 nucleotides) $T_nG$-repeat-like elements were not identified from any of the 48 distinct sets of ChIP–seq data for FOXA1, FOXM1, FOXJ2, FOXJ3, FOXQ1 and FOXS1, while FKHM ranked as one of the strongest motifs in many (Supplementary Table 1c). These results suggest that preference for $T_nG$-repeat-like sequence

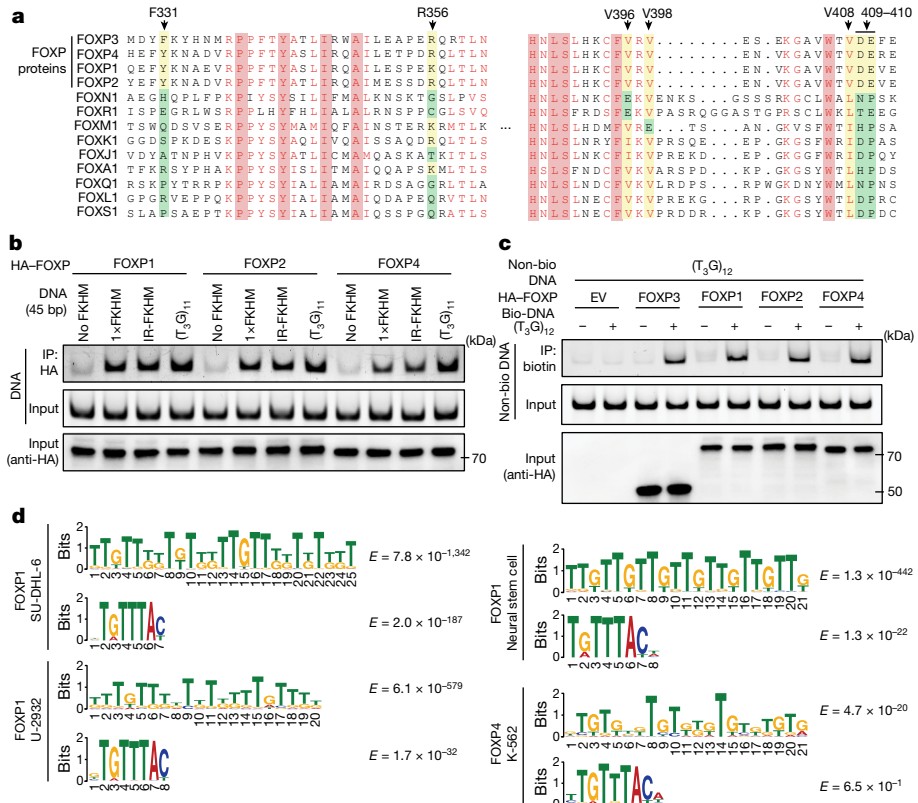

**Fig. 5 | T_nG microsatellite recognition is conserved among FOXP3 orthologues and paralogues. a**, Sequence alignment of forkhead TFs. Residues equivalent to the key interface residues in mouse FOXP3 (arrow on top) were highlighted in yellow (when similar to the mouse residues) or green (when dissimilar). **b**, The DNA-binding activity of FOXP1, FOXP2 and FOXP4. HA-tagged FOXP1, FOXP2 and FOXP4 were transiently expressed in HEK293T cells and purified by anti-HA immunoprecipitation. Equivalent amounts of the indicated DNAs (all 45 bp) were added to FOXP1/2/4-bound beads and further purified before analysis using gel analysis (SybrGold). **c**, The DNA-bridging activity of FOXP1, FOXP2 and FOXP4. Experiments were performed as described in Fig. 3c using HEK293T lysate expressing HA-tagged FOXP TFs. **d**, De novo motif analysis of FOXP1 and FOXP4 ChIP–seq peaks from a published database[38,39]. The comprehensive list and their references are provided in Supplementary Table 1c.

and ladder-like assembly are conserved properties of FOXP3 paralogues and orthologues, but may not be shared among all forkhead TFs.

## Discussion

In summary, our findings show a mode of TF–DNA interaction that involves TF homomultimerization and DNA bridging. After binding to T_nG repeats, FOXP3 forms a ladder-like multimer, in which FOXP3 uses two DNA molecules as scaffolds to facilitate cooperative multimeric assembly. That is, the first set of FOXP3 molecules (possibly a dimer or two dimers with an 8 bp spacing) that bridge DNA would help to recruit additional FOXP3 rungs, which would in turn stabilize the bridged DNA architecture and subsequent rounds of FOXP3 recruitment. Such cooperative assembly enables FOXP3 to preferentially target long repeats of T_nG rather than spurious sequences containing a few copies of T_nG. The DNA-bridging activity also implicates FOXP3 as a class of TF that can directly mediate architectural functions, which may explain the recently observed role of FOXP3 in chromatin loop formation[12,13].

Regarding how we can reconcile the ladder-like assembly of FOXP3 on T_nG repeats and the previously reported head-to-head dimeric structure on IR-FKHM or related sequences[22], much remains to be investigated. In contrast to the ladder-like multimerization, cellular evidence for the head-to-head dimerization is currently limited based on the available FOXP3 ChIP or CNR-seq data. Moreover, our new data showed that previously reported mutations that disrupt the head-to-head dimerization also affected the ladder-like multimerization, further limiting

the ability to probe the physiological functions of the head-to-head dimerization. Nevertheless, given that head-to-head dimerization is unique to FOXP3, while the ladder-like multimerization is shared among all four FOXP TFs, we speculate that both forms exist in cells and carry out distinct functions depending on the sequence of the bound DNA. For example, DNA bridging would be a unique consequence of the ladder-like assembly, not shared with the head-to-head dimer, while the head-to-head dimerization may enable the recruitment of certain cofactors using the unique surface created by the dimerization. This fits the previous microscopy analysis in which FOXP3 was found in two distinct types of nuclear clusters associated with different cofactors[16]. Together, these findings suggest that FOXP3 is a versatile TF that can interpret a wide range of sequences by assembling at least two distinct homomultimeric structures.

Our findings also implicate functional roles of microsatellites in FOXP TF-mediated transcription regulation. While widely used as genetic tracing markers due to their high degrees of polymorphism, reports of the biological functions of microsatellites[30,31], besides their well-known pathogenic roles[32–35], remain sparse[36,37]. Our finding of the T_nG repeat recognition by FOXP3 and other members of the FOXP family raises the question of whether microsatellites have greater and more direct roles in transcriptional regulation than previously thought. This also prompts speculation that microsatellite polymorphism may contribute to a broad spectrum of diseases through FOXP TF dysregulation, such as autoimmunity through FOXP3, neurodevelopmental disorders through FOXP1, speech and language impairments through FOXP2, and heart and hearing defects through FOXP4.

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

## Methods

### Mice

C57BL/6N mice, sourced from Taconic Biosciences and overseen by Harvard Medical Area (HMA) Standing Committee on Animals, were housed in an individually ventilated cage system at the specific-pathogen-free New Research Building facility of Harvard Medical School. The mice were maintained in a controlled environment with a temperature of 20–22 °C, humidity of 40–55% and under a 12 h–12 h light–dark cycle. The spleens of around 12–14 week old female C57BL/6 mice were isolated for the study.

### Naive CD4$^+$ T Cells

Cells were isolated using the Naive CD4$^+$ T Cell Isolation Kit (Miltenyi Biotec, 130-104-453) according to the manufacturer's instructions and maintained in complete RPMI medium (10% FBS heat-inactivated, 2 mM L-glutamine, 1 mM sodium pyruvate, 100 μM NEAA, 5 mM HEPES, 0.05 mM 2-ME).

### HEK293T and A549 cells

HEK293T cells (purchased from ATCC (CRL-11268)) and A549 cells (gift from S. Weiss) were maintained in DMEM (high glucose, L-glutamine, pyruvate) with 10% fetal bovine serum and 1% penicillin–streptomycin.

### EL4 cells

EL4 cells (gift from the C.B. laboratory) were cultured in DMEM (high glucose, L-glutamine, pyruvate) supplemented with 10% fetal bovine serum, ranging from $1 \times 10^5$ to $1 \times 10^6$ cells per ml.

### Plasmids

Mouse FOXP3 plasmids were generated as previously described[22]. For mammalian expression plasmids, HA-tagged mouse *FOXP3* coding sequence was inserted into the pcDNA3.1+ vector between the KpnI and BamHI sites. All FOXP3 mutations, including R356E, V396E, V398E, V408M and 409-411AAA, were generated by site-directed mutagenesis using Phusion High Fidelity (New England Biolabs) DNA polymerases. For retroviral packaging plasmids, HA-tagged mouse *FOXP3* coding sequence was inserted into the MSCV-IRES-Thy1.1 vector.

For mammalian expression plasmids of FOXP3 orthologues from *H. sapiens*, *O. anatinus* and *D. rerio*, the respective *FOXP3* coding sequence with overhangs of pcDNA vector was synthesized by IDTDNA and then assembled using the NEBuilder HiFi DNA Assembly Cloning Kit (NEB, 5520G). FOXP3 paralogue (FOXP1, FOXP2 and FOXP4) mammalian expression plasmids were made in the same way. Other forkhead TFs, such as FOXA1, FOXM1, FOXQ1 and FOXS1, were gifts from the S. Koch laboratory[40] through Addgene.

### DNA oligos

Single-stranded DNA (ssDNA) oligos were synthesized by IDTDNA. Double-stranded DNA (dsDNA) oligos for the electrophoretic mobility shift assay (EMSA) assay, pull-down assay and DNA-bridging assay were annealed from single-stranded, complementary oligos. After briefly centrifuging each oligonucleotide pellet, ssDNAs were dissolved in the annealing buffer (10 mM Tris-HCl pH 7.5, 50 mM NaCl). Complementary ssDNAs were then mixed together in equal molar amounts, heated to 94 °C for 2 min and gradually cooled down to room temperature. For dsDNA in cryo-EM analysis, high-performance-liquid-chromatography-purified single-stranded, complementary oligos were purchased from IDTDNA. After annealing, dsDNA was further purified by size-exclusion chromatography (SEC) on Superdex 75 Increase 10/300 (GE Healthcare) columns in 20 mM Tris-HCl pH 7.5, 150 mM NaCl. Biotin-labelled ssDNA oligos were synthesized by IDTDNA and then dissolved in annealing buffer (10 mM Tris-HCl pH 7.5, 50 mM NaCl). Complementary, biotin-labelled ssDNAs were then mixed together in equal molar amounts, heated to 94 °C for 2 min and gradually cooled down to room temperature. The sequences of all of the DNA oligos used are provided in Supplementary Table 2b.

### Protein expression and purification

All recombinant proteins in this paper were expressed in BL21(DE3) at 18 °C for 16–20 h after induction with 0.2 mM IPTG. Cells were lysed by high-pressure homogenization using the Emulsiflex C3 (Avestin) system. All proteins are from the *M. musculus* sequence, unless mentioned otherwise. FOXP3(ΔN) (residues 188–423) was expressed as a fusion protein with an N-terminal His$_6$–NusA tag. After purification using Ni-NTA agarose, the protein was treated with HRV3C protease to cleave the His$_6$–NusA-tag and was further purified through a series of chromatography purification using the HiTrap Heparin (GE Healthcare), Hitrip SP (GE Healthcare) and Superdex 200 Increase 10/300 (GE Healthcare) columns. The final SEC was performed in 20 mM Tris-HCl pH 7.5, 500 mM NaCl, 2 mM DTT. NFAT1 protein (residues 394–680) was also expressed as a fusion protein with an N-terminal His$_6$–NusA tag. After purification using Ni-NTA agarose, the His$_6$–NusA-tag was removed using the HRV3C protease and was further purified by SEC on the Superdex 75 Increase 10/300 (GE Healthcare) column in 20 mM Tris-HCl pH 7.5, 500 mM NaCl, 5% glycerol, 2 mM DTT. His$_6$–MBP-fused FOXP3(ΔN) variants were purified using the Ni-NTA affinity column and Superdex 200 Increase 10/300 (GE Healthcare) SEC column in 20 mM Tris-HCl pH 7.5, 500 mM NaCl, 2 mM DTT.

### MBP–FOXP3(ΔN) PD-seq

Mouse EL4 genomic DNA was isolated using the Qiagen Blood & Cell Culture DNA Kit (Qiagen, 13343). The purified genomic DNA was then fragmented to about 100–200 bp using DNase I (Zymo Research, E1010) in the digestion buffer (50 mM NaCl, 20 mM Tris-HCl PH 7.5, 1.5 mM MgCl$_2$) (for a 200 μl system, 50 μg genomic DNA was treated with 8 μl DNase I for around 3–4 min to obtain about 100–200 bp DNA fragments). The digested genomic DNA was then purified using the QIAquick Nucleotide Removal Kit (Qiagen, 28306) and used as an input for the PD-seq.

Purified MBP-tag or MBP–FOXP3(ΔN) protein was incubated with the input DNA fragments in the incubation buffer (20 mM Tris-HCl pH 7.5, 100 mM NaCl, 1.5 mM MgCl$_2$) for 20 min at room temperature and then processed for MBP pull-down using amylose resin (New England Biolabs). The bound DNA was recovered using proteinase K (New England Biolabs) and purified using the QIAquick Nucleotide Removal kit (Qiagen). The sequencing libraries were made using the NEBNext Ultra II DNA Library Prep Kit (Illumina) according to the manufacturer's instructions and submitted to Novogene for paired-end 150 bp NGS.

### Nucleosome PD-seq

Mouse EL4 cells were lysed using a hypotonic buffer (20 mM Bis-Tris pH 7.5, 0.05% NP-40, 1.5 mM MgCl$_2$, 10 mM KCL, 5 mM EDTA, 1× mammalian protease inhibitor) and the nuclear fraction was isolated by centrifuging at 4 °C and 2,500 rpm for 10 min. The isolated nuclear fraction was then digested with micrococcal nuclease (Thermo Fisher Scientific, 88216) for 1 h at 4 °C to fragment the chromatin into individual nucleosomes. The lysate was then centrifuged at 4 °C and 13,000 rpm for 10 min. The cleared lysate containing the nucleosomes was incubated with purified MBP-tag or MBP–FOXP3(ΔN) protein (1 μM) for 1 h at 4 °C and then processed for MBP pull-down using amylose resin (New England Biolabs). After treatment with proteinase K (New England Biolabs), the final nucleosomal DNAs were recovered using QIAquick Nucleotide Removal kit (Qiagen) and used for library preparation. The libraries were made using the NEBNext Ultra II DNA Library Prep Kit (Illumina) according to the manufacturer's instructions and submitted to Novogene for paired-end 150 bp NGS.

## MBP–FOXP3(ΔN) pull-down assay

Purified MBP–mFOXP3(ΔN) protein (0.4 μM) was incubated with 0.1 μM DNA in incubation buffer for 20 min. The FOXP3–DNA mixture was then incubated with 25 μl amylose resin (New England Biolabs) for 30 min with rotation at room temperature. The bound DNA was recovered using proteinase K (New England Biolabs), purified using the QIAquick Nucleotide Removal kit (Qiagen) and analysed on 10% Novex TBE gels (Invitrogen). DNA was visualized by Sybr Gold staining. The expression of MBP–FOXP3(ΔN) was validated by western blotting using mouse MBP tag antibodies (Cell Signaling Technology, 8G1, 2396, 1:2,000).

## HA–FOXP3 pull-down assay

HEK293T cells were transfected with pcDNA encoding HA-tagged FOXP3 (wild-type or mutants). After 48 h, cells were lysed using RIPA buffer (10 mM Tris-HCl pH 8.0, 1 mM EDTA, 1% Triton X-100, 0.1% sodium deoxycholate, 0.1% SDS, 140 mM NaCl and 1× proteinase inhibitor) and treated with benzonase (Millipore) for 30 min. The lysate was then incubated with anti-HA magnetic beads (Thermo Fisher Scientific) for 1 h. The beads were washed three times using RIPA buffer and incubated with DNA oligos for 20 min at room temperature. Bound DNA was recovered using proteinase K (New England Biolabs), purified using the QIAquick Nucleotide Removal kit (Qiagen) and analysed on 10% Novex TBE gels (Invitrogen). DNA was visualized by Sybr Gold staining.

## Nucleosome reconstitution and EMSA analysis

Nucleosome core particles were reconstituted with recombinant histone octamer H3.1 (Active motif) and DNAs as described previously[41]. In brief, 1 μM of TTTG repeats (144 bp), AAAG repeats (144 bp), TGTG repeats (144 bp) and DNA containing the 601 sequence (181 bp) were incubated with 1 μM of the histone octamer and were dialysed against 10 mM Tris-HCl PH 7.5, 1 mM EDTA, 2 mM DTT for 24 h. Nucleosomes (0.05 μM) were incubated with the indicated amount of FOXP3(ΔN) in the buffer (10 mM Tris-HCl pH 7.5, 50 mM NaCl, 1 mM EDTA and 2 mM DTT) for 30 min at 4 °C and analysed on 6% TBE gels (Life Technologies) at 4 °C. After staining with Sybr Gold stain (Life Technologies), Sybr Gold fluorescence was recorded using the iBright FL1000 (Invitrogen) system and analysed using the iBright analysis software.

## Biotin–DNA pull-down assay

HA–FOXP3 was transiently expressed in HEK293T cells as described above. Cells were lysed using RIPA buffer. The lysate was incubated with biotin–dsDNA (1 μM) for 1 h, and then with Streptavidin agarose beads (Thermo Fisher Scientific, 25 μl) for an additional 30 min. The beads were centrifuged and washed three times with RIPA buffer. Bead-bound protein was extracted using the SDS loading buffer and analysed by SDS–PAGE and western blotting using anti-HA (primary) antibodies (Cell Signaling, 3724S, 1:3,000) and anti-rabbit IgG-HRP (secondary) antibodies (Cell Signaling, 7074, 1:5,000).

## EMSA

DNA (0.05 μM) was mixed with the indicated amount of FOXP3 in buffer A (20 mM HEPES pH 7.5, 150 mM NaCl, 1.5 mM MgCl$_2$ and 2 mM DTT), incubated for 30 min at 4 °C and analysed on 3–12% gradient Bis-Tris native gels (Life Technologies) at 4 °C. After staining with Sybr Gold stain (Life Technologies), Sybr Gold fluorescence was recorded using the iBright FL1000 (Invitrogen) system and analysed using the iBright analysis software.

## Cross-linking analysis

Protein–protein cross-linking using BMOE (Thermo Scientific) was performed according to the product manual. In brief, 0.4 μM FOXP3(ΔN) was incubated with 0.05 μM DNAs at 25 °C for 10 min in 1× PBS, then BMOE was added to a final concentration of 100 μM. After incubation for 1 h at 25 °C, DTT (10 mM) was added to quench the cross-linking reaction. The samples were then analysed by SDS–PAGE and Krypton staining (Thermo Fisher Scientific).

## DNA-bridging assay

Biotin–DNA (bait, 0.1 μM) was incubated with Streptavidin agarose (25 μl, Thermo Fisher Scientific) in buffer B (20 mM Tris-HCl pH 7.5, 100 mM NaCl, 1.5 mM MgCl$_2$, 5 mM DTT) for 30 min by rotating the mixture at room temperature. Agarose beads were washed three times with buffer B and incubated with non-biotinylated DNA (prey, 0.1 μM) and purified FOXP3 protein (or HEK293T lysate expressing FOXP3). After incubation for 30 min with rotation, bead-bound DNA was recovered using proteinase K (New England Biolabs), purified using the QIAquick Nucleotide Removal kit (QIAGEN) and analysed on 10% Novex TBE gels (Invitrogen). DNA was visualized by Sybr Gold staining.

## Cryo-EM sample preparation and data collection

FOXP3(ΔN) was incubated with $(T_3G)_{18}$ DNA at a molar ratio of 8:1 in buffer B at room temperature for 10 min. The complex was then cross-linked using 0.5% glutaraldehyde for 10 min at room temperature before quenching with 1/10 volume of 1 M Tris-HCl pH 7.5 (for a final Tris concentration of 0.1 M). The FOXP3(ΔN)–DNA complex was then purified using the Superose 6 Increase 10/300 GL (GE Healthcare) column in 20 mM Tris-HCl pH 7.5, 100 mM NaCl, 2 mM DTT. The sample was concentrated to 1 mg ml$^{-1}$ (final for protein) and applied to freshly glow-discharged C-flat 300 mesh copper grids (CF-1.2/1.3, Electron Microscopy Sciences) at 4°C. The grids were plunged into liquid ethane after blotting for 5 s using the Vitrobot Mark IV (FEI) with a humidity setting of 100%. The grids were screened at the Harvard Cryo-EM Center and UMass Cryo-EM core facility using Talos Arctica microscope (FEI). The grids that showed a good sample distribution and ice thickness were used for data collection on the Titan Krios (Janelia Cryo-EM facility) system operated at 300 kV and equipped with a Gatan K3 camera. A total of 11,624 micrographs was taken at a magnification of ×81,000 with a pixel size of 0.844 Å. Each video comprised 60 frames at a total dose of 60 e$^-$ Å$^{-2}$. The data were collected in a desired defocus range of −0.7 to −2.1 mm.

## Cryo-EM data processing and structure refinement

Data were processed using cryoSPARC (v.4.2.0)[42] and RELION (v.4.0.1)[43,44]. The dose-fractionated videos were motion corrected using MotionCor2[45]. The contrast transfer function was estimated using CTFFIND (v.4.1)[46]. Particles were picked using the auto pick function in RELION[47]. A total of 4,201,166 raw particles was transferred to cryoSPARC for 2D classification. In total, 1,009,168 particles from selected 2D classes were used for ab initio reconstruction, in which they were divided into six ab initio classes. A total of 317,175 particles from class 1 was then refined to a final resolution of 3.7 Å with non-uniform refinement. To improve the local resolution, we performed local refinement using a mask covering the central FOXP3 tetramer, and obtained a map at a resolution of 3.3 Å. For structure refinement, a previous crystal structure of a FOXP3(ΔN) monomer bound to DNA (PDB: 7TDX) was docked into the EM density map from global refinement using UCSF Chimera[48]. A total of ten copies of FOXP3(ΔN) monomers were located for the global refinement map. For the mask-focused local refinement map, four copies of FOXP3(ΔN) monomers in complex with DNA were docked. Subsequently, the decamer and tetramer models were built manually against the respective density map using COOT[49], and refined using phenix.real_space_refine[50]. The structure validation was performed using MolProbity[51] from the PHENIX package. The curve representing model versus full map was calculated, based on the final model and the full map. The statistics of the 3D reconstruction and model refinement are summarized in Extended Data Table 1. All molecular graphics figures were prepared using PyMOL (Schrödinger) and UCSF Chimera[48]. All software used for cryo-EM data processing and model building was installed and managed by SBGrid[52].

## Negative-stain EM

FOXP3(ΔN) (0.4 μM) was incubated with DNA (0.05 μM) in buffer B at room temperature for 10 min. The samples were diluted tenfold with buffer A, immediately adsorbed to freshly glow-discharged carbon-coated grids (Ted Pella) and stained with 0.75% uranyl formate as described previously[53]. Images were collected using the JEM-1400 transmission electron microscope (JEOL) at ×50,000 magnification.

## De novo motif analysis of FOXP3-occupied sites in vitro and in vivo

FoxP PD-seq data were mapped to mm10 using Bowtie2[54] and sorted using samtools[55]. Peaks were called using MACS2[56] with either input or MBP pull-down as controls. The default settings were used for peak calling. De novo motif analysis was performed using MEME-ChIP[57] and STREAM[58] with the minimum and maximum motif lengths set at 6 and 30 nucleotides, respectively.

FOXP3 CNR-seq and ChIP–seq data[14] were mapped to mm10 using Bowtie2[54]. Peaks were called using MACS2[56]. Bedtools was used to obtain the CNR-seq consensus ($n$ = 1,372) and union ($n$ = 9,062) peaks between previously reported CNR peaks[12,14]. Motif analysis was performed as described above. To independently validate the results, similar motif analysis was repeated using different ChIP–seq data[23,24], which were mapped to the mm10 genome using Bowtie2. Peaks were called using HOMER with an input control[22] and were ranked on the basis of the signal intensity using samtools[55]. The top 5,000 overlapping FOXP3 ChIP–seq peaks were calculated by bedtools using a 50% reciprocal overlap criterion. FOXP3-negative open chromatin regions were derived from all observed $T_{reg}$ cell open chromatin regions[27]. Intersections and non-overlapping genomic features were extracted using the bedtools[59] intersect functionality and were processed for the motif analysis as above. The versions and parameters for software used above have been uploaded to GitHub (https://github.com/DylannnWX/Hurlab/tree/main/Foxp3_manuscript).

## Genome-wide analysis of $T_nG$-repeat-like elements

FIMO[60] was used to identify $T_nG$-repeat-like elements. The $T_nG$-repeat-like motif identified from the MEME-ChIP analysis of the overlap of previously reported CNR peaks[12,14] (Supplementary Table 1b) was used as a query motif, and a search was performed against the human (GrCh38), mouse (GrCm38) and Zebrafish (GrCz11) genomes. The default $P$-value cutoff ($P$ = 0.05) was used. FIMO outputs of all regions that match the query motif were converted to the .bed file format, and the overlapping $T_nG$ regions from FIMO outputs were combined into a single region using the bedtools merge function.

## Comparison between FOXP3 CNR union peaks with and without $T_nG$-repeat-like elements

FIMO[60] was used as described above to identify $T_nG$-repeat-containing peaks from the union peaks of previously reported CNR peaks[12,14] ($n$ = 9,062). Out of the 9,062 peaks, 3,301 peaks showed at least one $T_nG$ region lower than the default $P$-value cut-off ($P$ = 0.05), and were classified as $T_nG$-containing peaks. The non-$T_nG$-containing peaks were then calculated using bedtools peak subtraction with intersect -v. Genomic feature analysis was performed using ChIPseeker[61]. To compare H3K4me3, H3K27ac and ATAC signal intensity, H3K4me3 and H3K27ac ChIP–seq and ATAC–seq data[23] were mapped to the mm10 genome using Bowtie2[54] and the intensity was calculated within 2 kb upstream and downstream of the FOXP3 CNR peak summits using Deeptools[62] bamCoverage and Deeptools computeMatrix. The versions and parameters for the software used above have been uploaded to GitHub (https://github.com/DylannnWX/Hurlab/tree/main/Foxp3_manuscript).

## Motif analysis of other forkhead TFs

Peak bed files for FOXP1, FOXP2, FOXP4, FOXJ2, FOXJ3, FOXA1, FOXM1, FOXS1 and FOXQ1 were downloaded from ChIP-Atlas (http://chip-atlas.org/) and converted to fasta files using bedtools[59] getfasta. The individual fasta file was then processed for de novo motif analysis using MEME-ChIP[57] with the minimum and maximum motif lengths set at 6 and 30 nucleotides, respectively. The results are summarized in Supplementary Table 1c.

## CD4+ T cell isolation and retroviral transduction

Naive CD4+ T cells were isolated by negative selection from mouse spleens using the isolation kit (Miltenyi Biotec) according to the manufacturer's instruction. The purity was estimated to be >90% as measured by PE anti-CD4 (BioLegend, 100408, 1:1,000) staining and FACS analysis. Naive CD4+ T cells were then activated with anti-CD3 (BioLegend, 100340, 1:500 dilution to 5 μg ml⁻¹), anti-CD28 (BioLegend, 102116, 1:500 dilution to 5 μg ml⁻¹) and 50 U ml⁻¹ of IL-2 (Peprotech) in complete RPMI medium (10% FBS heat-inactivated, 2 mM L-glutamine, 1 mM sodium pyruvate, 100 μM NEAA, 5 mM HEPES, 0.05 mM 2-ME). The activation state of T cells was confirmed by increased cell size and CD44 (BioLegend) expression using FACS. After 48 h, cells were spin-infected with retrovirus-containing supernatant from HEK293T cells that were transfected with retroviral expression plasmids (Empty MSCV-IRES-Thy1.1 vector, wild-type FOXP3 and mutations encoding vectors) and cultured for about 2–3 days in complete RPMI medium with 100 U ml⁻¹ of IL-2.

## FOXP3 transcriptional activity assay in CD4+ T cells

FOXP3 transcriptional activity was measured by the levels of two known target genes, *CD25* and *CTLA4*, and the FOXP3 expression marker Thy1.1. FOXP3-transduced CD4+ T cells were stained with antibodies targeting the cell-surface antigens CD25 (BioLegend, 102022, 1:1,000) and Thy1.1 (BioLegend, 202520, 1:1,000) on day 2 after retroviral infection. The level of CTLA4 was measured by intracellular staining using anti-CTLA4 (BioLegend, 106311, 1:1,000) antibodies and the Transcription Factor Staining Buffer Set (eBioscience) on day 3 after retroviral infection. Flow cytometry data were analysed using FlowJo software and presented as plots of mean fluorescence intensity of CD25 and CTLA4 in cells grouped into bins of Thy1.1 intensity, which is the expression marker for FOXP3. Each result is representative of three independent experiments.

## FOXP3 ChIP–seq analysis

FOXP3 ChIP–seq was conducted using CD4+ T cells according to a published procedure[16]. Activated CD4+ T cells that had been transduced with wild-type or mutant *FOXP3* were sorted based on Thy1.1 reporter expression. For each sample (5 × 10⁶ cells), cross-linking was achieved with 1% formaldehyde for 10 min. Subsequently, the cells were lysed on ice using RIPA buffer (10 mM Tris-HCl pH 8.0, 1 mM EDTA, 1% Triton X-100, 0.1% sodium deoxycholate, 0.1% SDS, 140 mM NaCl and 1× proteinase inhibitor). Chromatin fragmentation was achieved using an AFA Focused-ultrasonicator (Covaris M220) for 30 min (5% duty cycle, 140 W max power, 200 cycles per burst), resulting in DNA fragments ranging from 100 to 200 bp. The sheared material underwent centrifugation for 10 min at 13,000 rpm at 4 °C to clear the solution. The cleared material was then processed for immunoprecipitation overnight with anti-HA-tag antibodies (Cell Signaling, 3724) at 4 °C, and protein G beads (Active motif, 53014) were added for an additional 2 h. The beads were sequentially washed with various buffers: RIPA wash buffer (0.1% SDS, 0.1% sodium deoxycholate, 1% Triton X-100, 1 mM EDTA, 10 mM Tris-HCl pH 8.0, 150 mM NaCl), RIPA 500 wash buffer (0.1% SDS, 0.1% sodium deoxycholate, 1% Triton X-100, 1 mM EDTA, 10 mM Tris-HCl pH 8.0, 500 mM NaCl), LiCl wash buffer (10 mM Tris-HCl, pH 8.0, 250 mM LiCl, 0.5% Triton X-100, 0.5% sodium deoxycholate) and Tris buffer (10 mM Tris-HCl, pH 8.5). The chromatin was eluted from the beads using

elution buffer (1× TE, pH 8.0, 0.1% SDS, 150 mM NaCl, 5 mM DTT). After elution, the DNA was treated with 1 µg DNase-free RNase (Roche) for 30 min at 37 °C, followed by treatment with proteinase K (Roche) for at least 4 h at 63 °C to reverse the cross-links. The reverse-cross-linked DNA was then purified using SPRI beads (Beckman, B23318). Subsequent steps, including end repair, A-base addition, adaptor ligation and PCR amplification, were performed to prepare the ChIP–seq library for each sample. The libraries were generated using the NEBNext Ultra II DNA Library Prep Kit (Illumina) according to the manufacturer's instructions and submitted to Novogene for paired-end 150 bp NGS.

## mRNA-seq analysis

mRNA-seq was conducted using CD4$^+$ T cells. Activated CD4$^+$ T cells that had been transduced with wild-type or mutant *FOXP3* were sorted on the basis of Thy1.1 reporter expression. For each sample, $1 \times 10^6$ cells were sorted and processed for total RNA extraction using TRIzol reagent and the Direct-zol RNA Miniprep Kit. Quality control and the construction of mRNA-seq libraries were performed by Novogene. The NEB Next Ultra II kit and the non-directional mRNA approach with the poly(A) pipeline were used. The libraries were subsequently sequenced using the Illumina NovaSeq 6000 instrument, generating paired-end reads with a length of $2 \times 150$ bp, resulting in about 30 million reads per sample. Raw sequence files were subjected to pre-processing using Trimmomatic v.0.36 to remove Illumina adaptor sequences and low-quality bases. Trimmed reads were then aligned to the mouse genome (UCSC mm10) using bowtie2/2.3.4.3. For gene read counting, HTseq-count (v.0.12.4) was used. Normalization of gene counts and differential analysis were performed using DESeq2 (v.5). Heat maps were created using Pheatmap.

## Chromatin contact analysis

Hi-C- and PLAC-seq datasets were downloaded from the Gene Expression Omnibus (GSE217147)[12], and the list of $T_{reg}$ cell enhancer–promoter loops (EPLs) was obtained from a previous study[13]. All .hic files were converted to .cool files using hic2cool, and all .cool files were decompressed into .txt files using the cooler dump --join function. These decompressed files were loaded as Python pandas dataframes. All possible bins in .cool files were converted to bed file formats, and intersected with $T_nG$-containing or $T_nG$-absent CNR union peaks using the bedtools intersect -wa function to acquire the bins that contain $T_nG$ bins and non-$T_nG$ (N$T_nG$) bins. These bins were used as anchors to filter raw .cool files for contact pairs between $T_nG$–$T_nG$ (2$T_nG$), $T_nG$–N$T_nG$ ($T_nG$N$T_nG$) and N$T_nG$–N$T_nG$ (2N$T_nG$). These contact pairs were then filtered by (more than 5 in WT $T_{reg}$ cell Hi-C-seq) and (more than indicated threshold in FOXP3 PLAC-seq). A list of contact counts in Fig. 2f is provided in Supplementary Table 3.

The $P$ value of 2$T_nG$ pair enrichment was first calculated by getting the expected 2$T_nG$ pair counts in a given list of pairs assuming random distribution (number of contact pairs × proportion of all potential $T_nG$ bins$^2$). Then, this number was compared with the observed 2$T_nG$ pair counts using binomial distribution. The proportion of all potential $T_nG$ bins is 0.37, which matches the proportion of $T_nG$ CNR peaks out of all CNR peaks (3,301 out of 9,062). The $P$ value was the cumulated probability that the observed 2$T_nG$ pair counts happen by chance, and the alternative hypothesis, if the $P$-value is low, indicates the probability that in 2$T_nG$ pair is enriched in the given list of contact.

To compare Hi-C/PLAC-seq anchors (in mm9) to enhancer–promoter loop anchors (in mm10), the reference genomes of mm9 were lifted to mm10 using the UCSC genome browser to acquire the correlating bin coordinates in mm10, and their overlaps were analysed using the bedtools intersect function.

## T cell suppression assay

Isolated naive CD4$^+$ T cells were activated with anti-CD3 (BioLegend) and anti-CD28 (BioLegend) antibodies and 50 U ml$^{-1}$ of IL-2 (Pepro-tech) in complete RPMI medium. After 48 h, activated CD4$^+$ T cells were retrovirally transduced to express FOXP3 and were used as suppressors. In parallel, freshly isolated naive CD4$^+$ T cells were labelled with CellTrace CFSE (Invitrogen) and used as responders. CD3$^-$ T cells representing APC cells were also isolated using the isolation kit (Miltenyi Biotec) according to the manufacturer's instructions. For the suppression assay, the CFSE-labelled responder cells ($5 \times 10^4$ cells) were stimulated with APC cells ($10^4$ cells) and anti-CD3 (1 µg ml$^{-1}$) antibodies in 96-well round-bottom plates for 3 days in the presence or absence of FOXP3-transduced suppressor cells (at a responder-to-suppressor ratio of 2:1). The proliferation ratio of the responders was calculated as a function of CFSE dye dilution by FACS analysis.

## Statistics and reproducibility

Data in Figs. 1f–j, 2e, 3b–e, 4b,d–g and 5b,c and Extended Data Figs. 1g–l, 2a,h, 3b,c,e,f, 5c–e and 6a–e,g are representative of at least three independent experiments and each experiment was repeated independently with similar results.

## Reporting summary

Further information on research design is available in the Nature Portfolio Reporting Summary linked to this article.

## Data availability

Naked genomic DNA PD-seq, nucleosome PD-seq, *Foxp3* mRNA-seq and FOXP3 ChIP–seq data have been deposited at the Gene Expression Omnibus under accession code GSE243606. The structures and cryo-EM maps have been deposited at the PDB and the Electron Microscopy Data Bank under accession codes 8SRP and EMD-40737 for decameric FOXP3 in complex with DNA, and 8SRO and EMD-40736 for the central FOXP3 tetramer in a complex with DNA (focused refinement). Other research materials reported here are available on request.

## Code availability

All custom codes used in this project have been deposited at GitHub (https://github.com/DylannnWX/Hurlab/tree/main/Foxp3_manuscript). These include the processing of Deeptools matrix outputs, FIMO region to peak bed files and HiC/Cool data processing. All are standalone Jupyter Notebook instances. In each instance, detailed user instructions, example inputs and expected outputs were also included in this GitHub repository.

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

**Acknowledgements** We thank all of the members of the Hur laboratory, A. Rudensky and Y. Zhong for their discussions and feedback. This study was supported by the Modell fellowship to W.Z., NIH grants (R01AI180137, R01AI154653 and R01AI111784 to S.H. and R01AI165697 to C.B.) and the Howard Hughes Medical Institute (S.H.). Cryo-EM data were collected at the Cryo-EM facilities at Janelia, Harvard Medical School and University of Massachusetts Worcester.

**Author contributions** W.Z., F.L. and S.H. conceived and designed the project. W.Z. and F.L. performed all of the experiments. F.L. determined the structure. W.Z., X.W. and R.N.R. performed bioinformatic analysis. J.P. assisted experiments. C.B. and S.H. supervised bioinformatic analysis. S.H. supervised the overall project.

**Competing interests** The authors declare no competing interests.

**Additional information**
**Correspondence and requests for materials** should be addressed to Sun Hur.

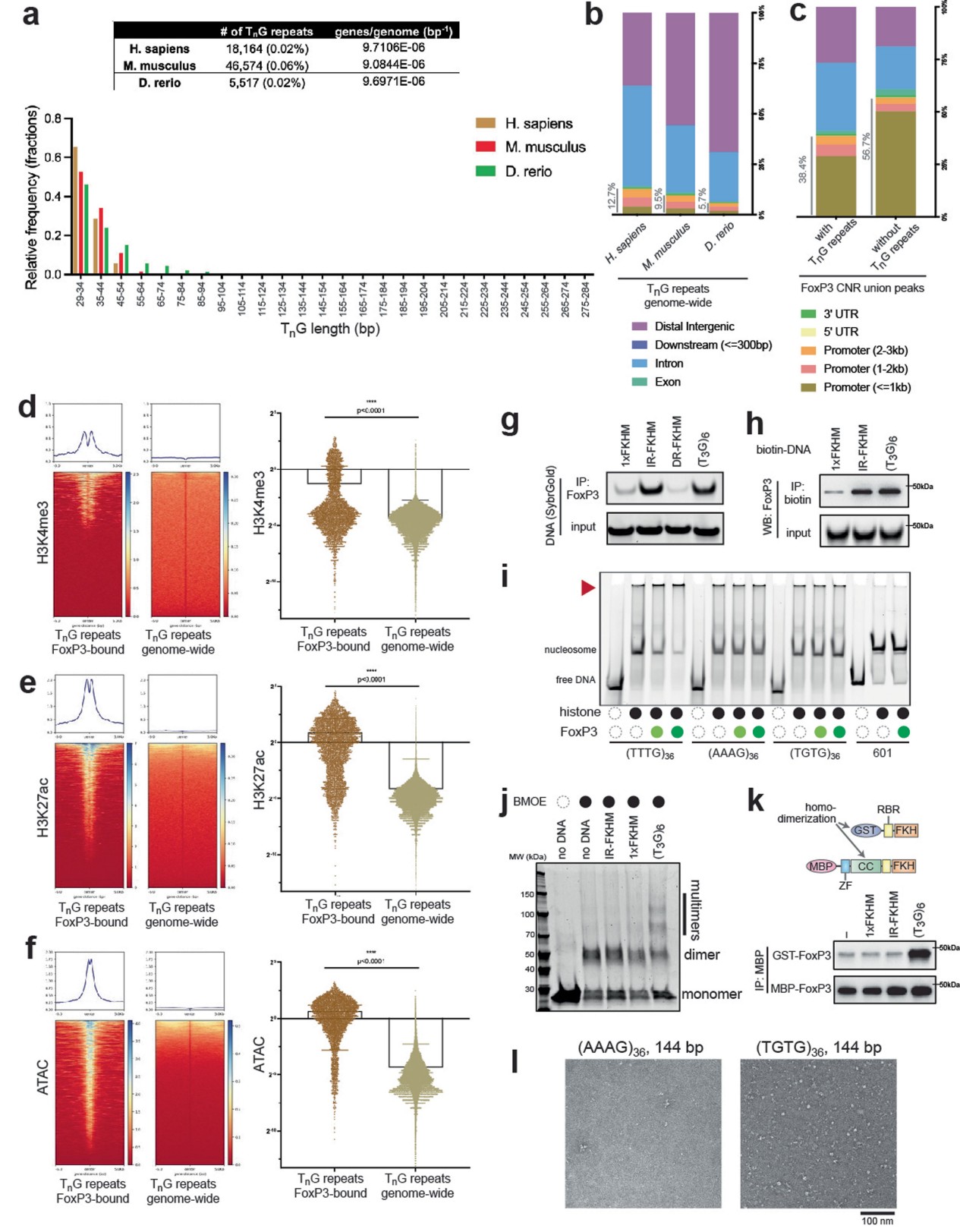

**Extended Data Fig. 1** | See next page for caption.

**Extended Data Fig. 1 | Analysis of $T_3G$ repeats in the genome and FoxP3 multimerization on $T_3G$ repeats.** a. $T_nG$ repeat-like sequences in the genomes of *H. sapiens*, *M. musculus* and *D. rerio*. Sequences that match the $T_nG$ repeat-like motif (29 nt motif from FoxP3 CNR overlap peaks, see Supplementary Table 1b) were identified using FIMO ($p$ = 0.05, see Methods). Genomic percentage of $T_nG$ repeat-like sequences (in parenthesis) was the number of $T_nG$ repeat regions multiplied by the average size of the repeats (31, 33 and 38 bp for *H. sapiens*, *M. musculus* and *D. rerio*, respectively), divided by the genome size (3.2, 2.7 and 1.4 billion bp, respectively). Below: length distribution of the TnG repeat-like sequences. Genes-to-genome size ratio was calculated by dividing the number of genes used in the feature annotation (31,074, 24,528 and 13,576 in *H. sapiens*, *M. musculus* and *D. rerio*) by the genome size. b. Distribution of $T_nG$ repeat-like sequences in the genomes of *H. sapiens*, *M. musculus* and *D. rerio* relative to Transcription Start Sites (TSSs). c. Distribution of CNR union peaks (union of Rudensky CNR peaks and Dixon CNR peaks, n = 9,062) relative to TSSs. CNR union peaks with and without $T_nG$ repeat-like sequences (n = 3,301 and 5,761, respectively) were identified using FIMO ($p$ = 0.05) as in (a). d-f. Comparison of (d) H3K4me3-ChIP, (e) H3K27ac-ChIP and (f) ATAC signal[23] around the $T_nG$ repeat-like sequences that overlap with FoxP3 CNR union peaks vs. those genome-wide in thymic Tregs (n = 4,837 peaks and 41,889 peaks respectively). See Extended Data Fig. 4a–c for pre-thymic Tregs, which showed that high levels of H3K4me3, H3K27ac and ATAC signals were maintained prior to FoxP3 expression. $T_nG$ repeats in the blacklist were removed. Right: ChIP/ATAC signal was averaged over +/− 500 bp around the $T_nG$ repeat-like sequences. Two-tailed unpaired t-tests. ****, $p$ < 0.0001. g. DNA sequence specificity of FoxP3 as measured by FoxP3 pull-down. HA-tagged, full-length FoxP3 was transiently expressed in HEK293T cells and purified by anti-HA IP. Equivalent amounts of indicated DNAs (30-31 bp) were added to FoxP3-bound beads and further purified by anti-HA IP prior to gel analysis. h. DNA sequence specificity of FoxP3 as measured by DNA pull-down. Equivalent amounts of biotinylated DNAs were mixed with FoxP3-expressing 293 T lysate and were subjected to streptavidin pull-down. Co-purified FoxP3 was analysed by anti-HA WB. i. FoxP3 binding to nucleosomal DNA as measured by native gel-shift assay. Indicated DNA was incubated with the histone octamer at 1:1 molar ratio (black circle), followed by incubation with FoxP3 (0.2 or 0.4 µM for light and dark green circles, respectively). Empty dotted circles indicate no histone or FoxP3. Sybrgold stain was used for visualization. With an increasing concentration of FoxP3, the intensity of the nucleosomal TTTG repeat decreased, while the signal in the gel well (red arrow) increased. Such changes were not observed with other DNAs. j. BMOE crosslinking of FoxP3$^{\Delta N}$ with and without DNA. FoxP3$^{\Delta N}$ can only form multimers on $(T_3G)_6$ DNA. k. Multimerization analysis of FoxP3, as measured by co-purification of FoxP3 with different tags. GST- and MBP-tagged FoxP3 were incubated together in the presence and absence of indicated DNA and were subjected to MBP pull-down, followed by WB analysis of GST-FoxP3 in eluate. Note that GST replaced the CC domain in FoxP3, disallowing hetero-dimerization between MBP-FoxP3 and GST-FoxP3. Thus, co-purification of these two proteins in the presence of $T_3G$ repeats suggests DNA sequence-dependent multimerization of the FoxP3 homodimer. l. Representative negative-stain EM images of FoxP3$^{\Delta N}$ in complex with $(AAAG)_{36}$ (left) and $(TGTG)_{36}$ (right).

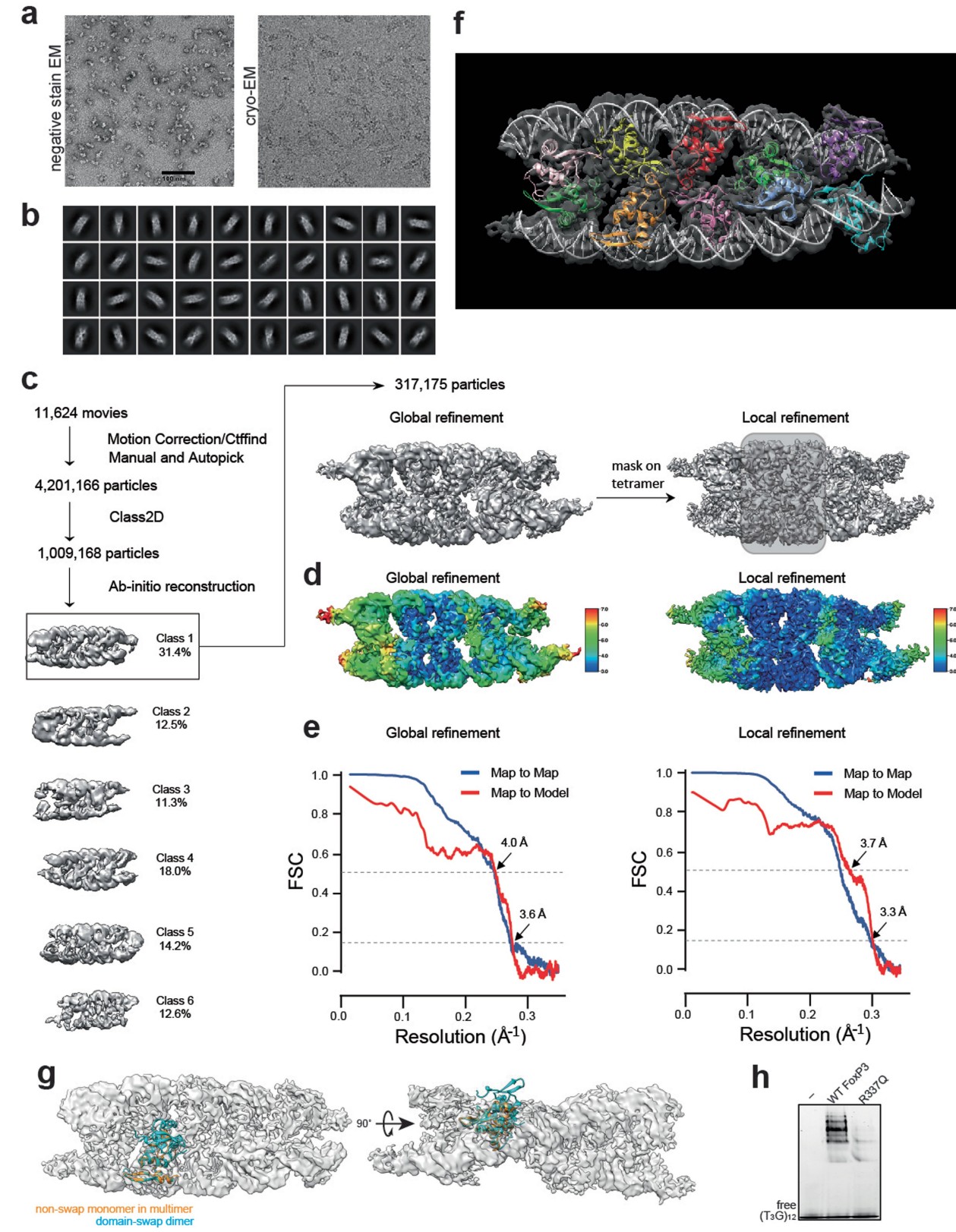

**Extended Data Fig. 2** | See next page for caption.

**Extended Data Fig. 2 | Cryo-EM structure of the FoxP3$^{\Delta N}$–(T$_3$G)$_{18}$ complex.**
a. Representative negative-stain EM (left) and cryo-EM images (right) of FoxP3$^{\Delta N}$ multimers on (T$_3$G)$_{18}$ DNA. b. 2D classes chosen for 3D reconstruction. c. Cryo-EM image processing workflow. See details in Methods. d. Local resolution for the maps of global refinement (left) and local refinement with a mask covering the central four subunits of FoxP3 (right). Local resolution was calculated by CryoSPARC. Resolution range was indicated according to the colour bar. e. Fourier shell correlation (FSC) curve for global refinement (left) and local refinement (right). Map-to-Map FSC curve was calculated between the two independently refined half-maps after masking (blue line), and the overall resolution was determined by gold standard FSC = 0.143 criterion. Map-to-Model FSC was calculated between the refined atomic models and maps (red line). f. Cryo-EM map and ribbon model of FoxP3$^{\Delta N}$ decamer in complex with two (T$_3$G)$_{18}$ DNAs (PDB: 8SRP, EMDB: 40737). DNA molecules are coloured grey. Individual FoxP3 monomers are coloured differently. g. Superposition of the domain-swap dimeric structure of FoxP3 (cyan, PDB:4WK8) onto any subunit of the FoxP3 multimeric structure (represented here by the orange subunit) by aligning the common portions of FoxP3 reveals that the domain-swap dimer is incompatible with the density map. h. Native gel shift analysis of MBP-tagged FoxP3$^{\Delta N}$ (WT or R337Q, 0.4 μM) with (T$_3$G)$_{12}$ DNA (0.05 μM). Note that R337Q induces domain-swap dimerization.

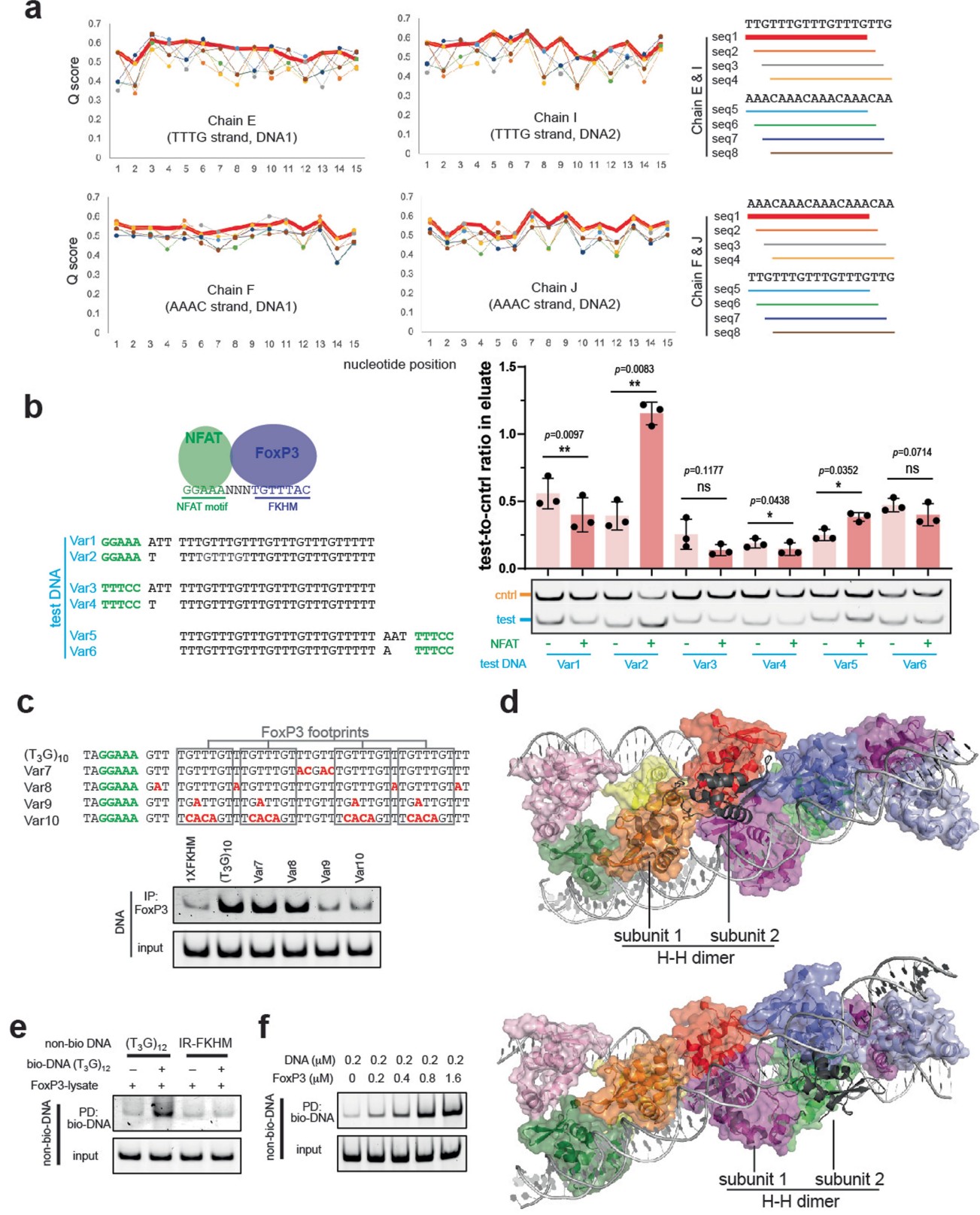

**Extended Data Fig. 3** | See next page for caption.

**Extended Data Fig. 3 | DNA sequence and conformational analysis.** a. DNA sequence assignment by Q-score analysis using mapQ. For each of the four DNA strands, eight possible sequence alignments (right) were tested against the local refinement map. The sequence alignment showing the highest overall score was highlighted with a thick red line. The best sequence alignments for all four DNA strands were consistent with each other and were used for cryo-EM reconstruction. b. Experimental validation of FoxP3–DNA registers using NFAT–FoxP3 cooperativity analysis. This assay utilizes the FoxP3 interaction partner NFAT, which assists FoxP3 binding to DNA only when their binding sites are 3 bp apart in one particular orientation (as in the schematic). To investigate FoxP3 footprints on $T_3G$ repeat DNA, we varied the position and orientation of NFAT consensus sequence (GGAAA, green) relative to $T_3G$ repeats (Var1-6), and performed FoxP3 pull-down. An internal control DNA (cntrl, harbouring the NFAT motif followed by FKHM with 3 nt gap) was used to normalize the test DNA (Var1-6) pull-down efficiency. Only DNA with a single nucleotide gap between GGAAA and TTTG (Var2) showed a positive effect of NFAT on the FoxP3–DNA interaction. This suggests that the most upstream FoxP3 subunit recognizes TGTTTGT. Two-tailed paired t-tests, comparing with and without NFAT.

$p < 0.005$ for **, $p < 0.05$ for * and $p > 0.05$ for ns. c. FoxP3 interaction with $(T_3G)_{10}$ variants. Variations in DNA sequence outside the FoxP3 footprints were tolerated (Var7 and Var8), but those within the footprints (Var9 and Var10) were not. d. Comparison of the inter-subunit interactions in FoxP3 decamer on $T_nG$ repeats vs head-to-head (H-H) dimer on IR-FKHM (PDB:7TDX). Superposition of the H-H dimer (dark grey) onto any of the ten subunits in the decamer structure showed distinct modes of inter-subunit interactions. Shown are two examples where subunit 1 of the H-H dimer was aligned to orange (top) or magenta (bottom) subunits of the decamer, showing that subunit 2 of the H-H dimer did not align with any of the decamer subunits. e. DNA bridging assay using FoxP3 expressed in 293 T cells. Biotinylated and non-biotinylated DNA (82 and 60 bp, respectively) were mixed at 1:1 ratio and further incubated with 293 T lysates expressing HA-tagged FoxP3, followed by streptavidin pull-down and gel analysis of non-biotinylated DNA by SybrGold staining. f. DNA bridging assay using an increasing concentration of purified FoxP3$^{\Delta N}$. 0.1 μM each of biotinylated and non-biotinylated $(T_3G)_{12}$ DNAs were used. FoxP3 concentrations are indicated at the bottom.

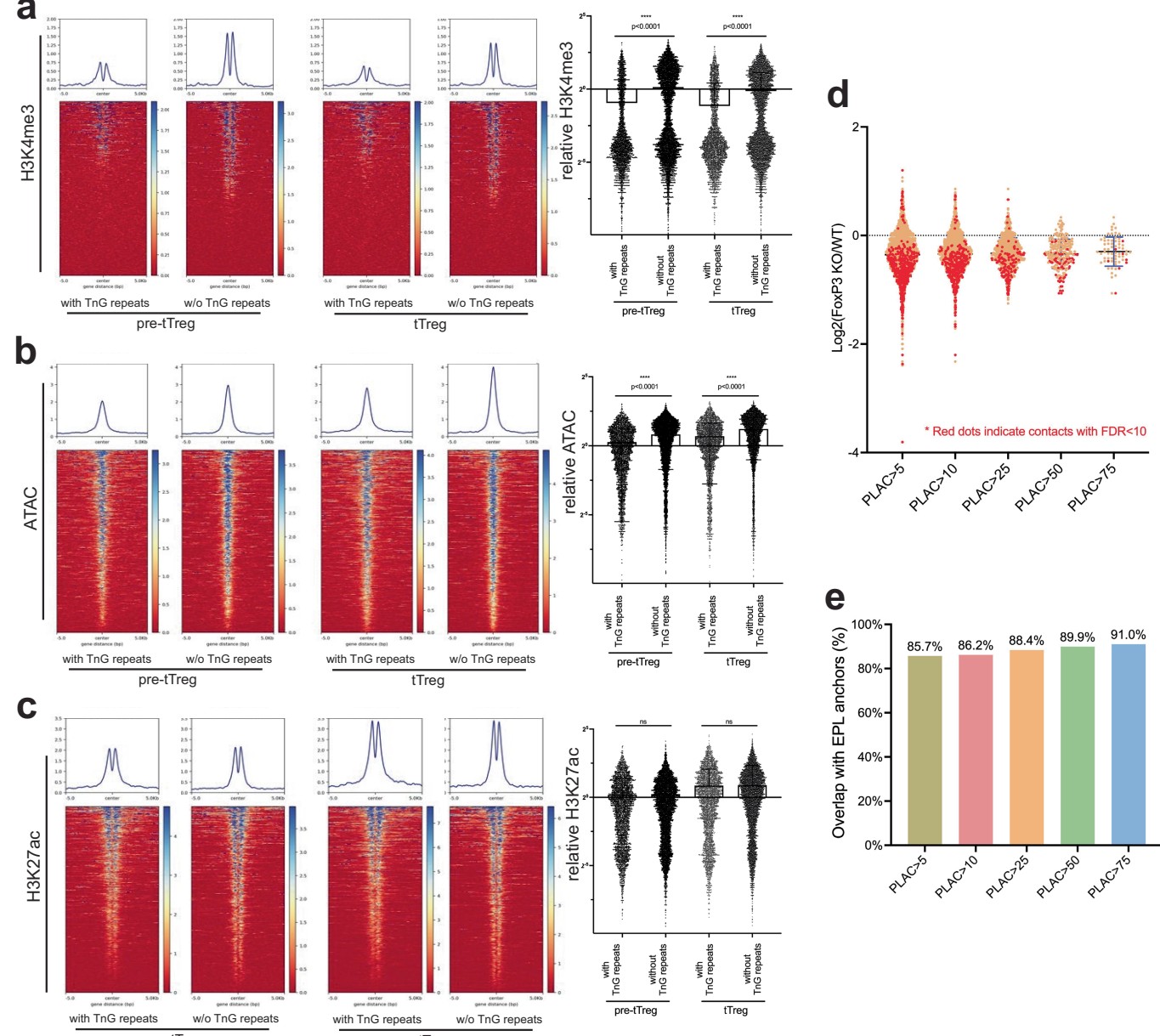

**Extended Data Fig. 4 | FoxP3 can bridge $T_nG$ repeat-containing sites in vivo.**
a-c. Comparison of (a) H3K4me3-ChIP, (b) ATAC and (c) H3K27ac-ChIP signal[23] around the CNR union peaks with and without $T_nG$ repeat-like sequences in pre-thymic Tregs (pre-tTregs) and thymic Tregs (tTregs) (n = 3,301 peaks and 5,761 peaks, respectively). Right: ChIP/ATAC signal was averaged over +/− 500 bp around the CNR peak summits. Two-tailed unpaired t-tests. ****, $p < 0.0001$.
d. FoxP3-dependence of the chromatin contacts at FoxP3-bound $T_nG$ anchors in Fig. 2f. FoxP3-bound $T_nG$ anchors were defined as anchors that overlap with FoxP3 CNR peaks with $T_nG$ repeat-like sequences. Contacts with frequency>5 in WT Treg HiC and connected by two $T_nG$ anchors were analysed with an increasing FoxP3 PLAC-seq count threshold. For each contact, log2 foldchange of HiC counts from WT to FoxP3 knock-out Treg were plotted. Contacts with FDR < 10 were coloured red. The majority of the $T_nG$–$T_nG$ contacts were less frequent in FoxP3 knock-out than in WT Tregs, although smaller fractions (10-15%) showed statistically significant FoxP3 dependence (FDR < 10), as previously reported[12]. n = 4365, 2559, 813, 204 and 60 anchors respectively. Mean ± SD were shown in black and blue lines. See also Supplementary Table 3. e. Fraction of the FoxP3-bound $T_nG$ anchors from Fig. 2f that overlap with previously published Treg EPL anchors[13].

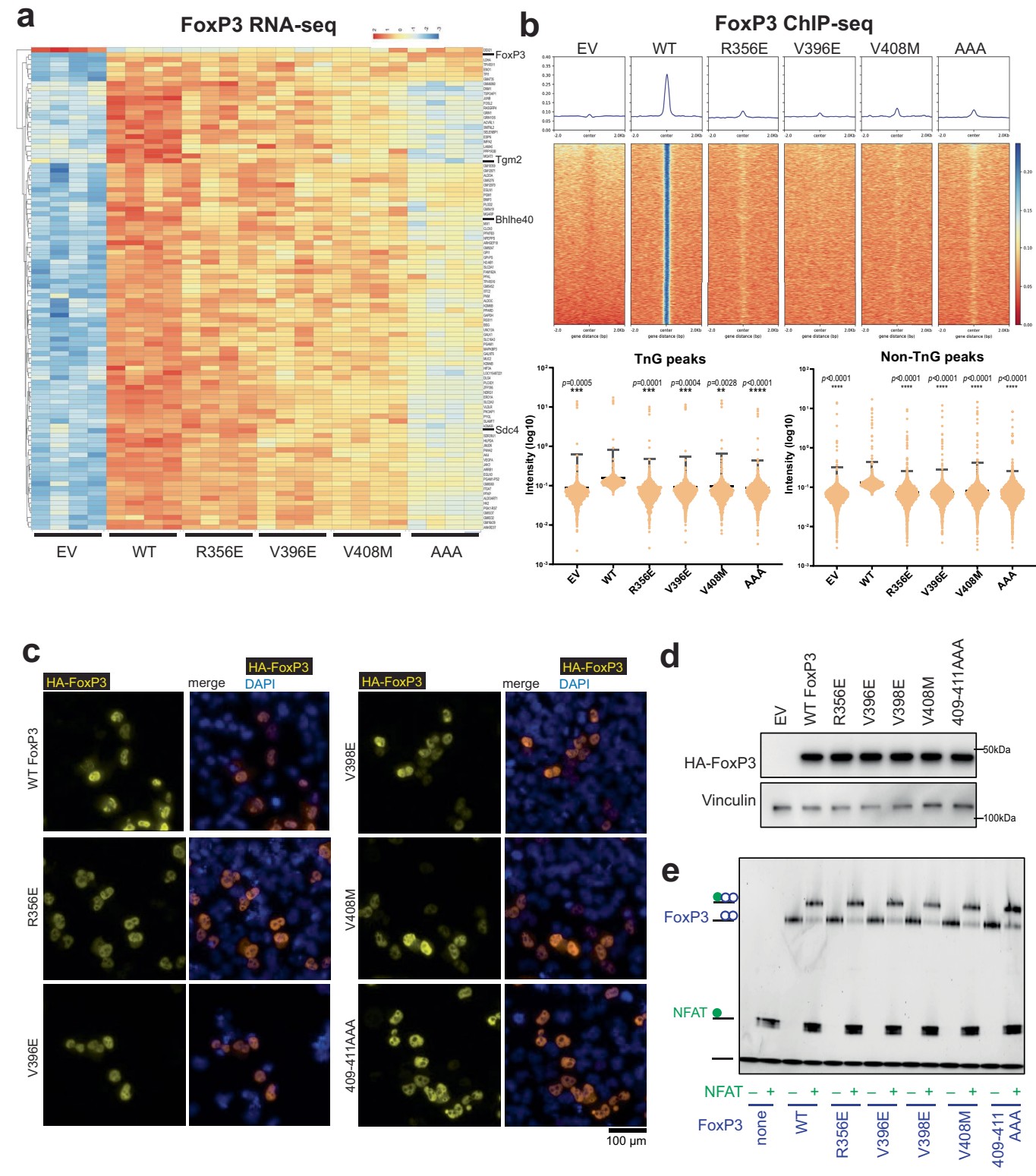

**Extended Data Fig. 5** | See next page for caption.

**Extended Data Fig. 5 | Characterization of intra-rung interface mutant FoxP3.** a. mRNA-seq heatmap analysis. CD4$^+$ T cells were transduced and sorted to express FoxP3 as in Fig. 3d and were subjected to mRNA-seq. Top 100 genes showing the most significant difference between WT FoxP3 and EV were chosen for the evaluation of individual mutants. All four mutants were impaired in transcriptional functions, albeit to varying extents. The level of FoxP3 was equivalent for WT and all mutants. A few genes previously reported to be FoxP3-dependent were indicated in larger fonts. Note that V398E was not tested due to its negative effect on NFAT binding in (e). b. ChIP-seq of HA-tagged FoxP3. Cells were transduced as in Fig. 3d and were subjected to anti-HA ChIP-seq. WT FoxP3 bound peaks were identified using MACS2 (n = 8,607, $p < 0.01$), and heatmaps of the ChIP signal were generated for each mutant at the WT peak locations. Below: averaged intensity of ChIP signal within 0.5 kb of the WT peak summit. Peaks with and without $T_nG$ repeats (n = 1,900 peaks and 6,707 peaks, respectively) were compared. Two-tailed paired t-tests, comparing mutants to WT. $p < 0.0001$ for ****, $p < 0.001$ for *** and $p < 0.005$ for **. c. Nuclear localization of WT FoxP3 and intra-rung interface mutants. HA-tagged FoxP3 was transiently expressed in A549 cells and was subjected to anti-HA immunofluorescent (yellow) analysis. Nuclei were shown with DAPI (blue) staining. d. Expression levels of WT FoxP3 and intra-rung mutants in A549 cells. e. Effect of the intra-rung mutations on the NFAT–FoxP3 interaction, as measured by native gel shift assay. FoxP3 (0.1 μM) was incubated with DNA harbouring IR-FKHM and the NFAT site (with a 3-bp gap as in Extended Data Fig. 3b, 0.05 μM). NFAT (0.1 μM) was added to the mixture to monitor formation of the ternary complex NFAT–FoxP3–DNA. Note that V398E showed slight but reproducible reduction in NFAT binding.

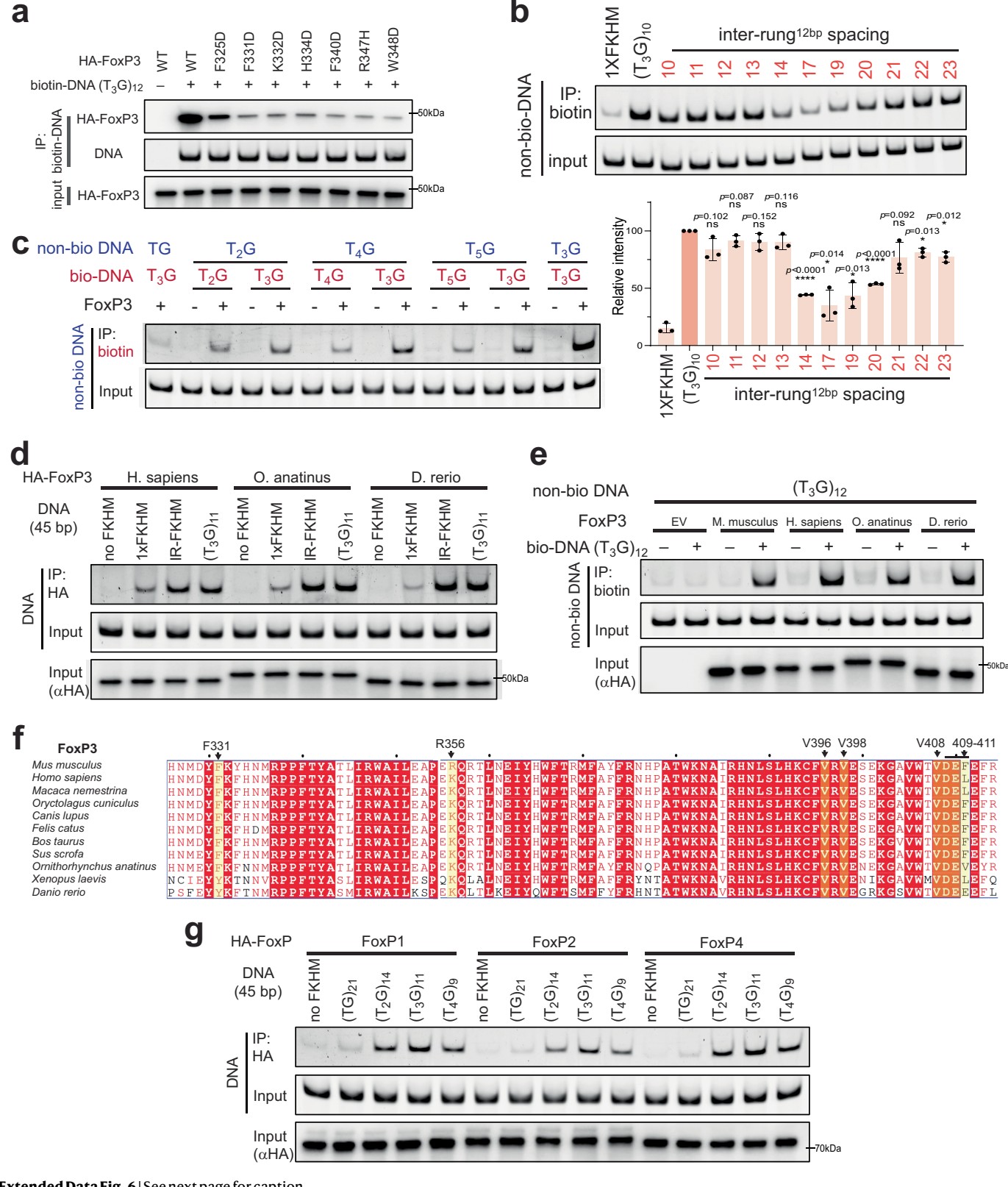

**Extended Data Fig. 6** | See next page for caption.

**Extended Data Fig. 6 | Multimerization on $T_nG$ repeats is conserved in FoxP3 orthologs and paralogs.** a. $T_nG$ repeats DNA-binding activity of FoxP3 with mutations in RBR. All seven RBR mutations previously shown to disrupt the head-to-head dimerization[22] disrupted $(T_3G)_{12}$ binding. b. Effect of inter-rung[12bp] spacing variations on FoxP3-mediated DNA bridging. Non-biotinylated DNAs in Fig. 4d were mixed with biotinylated $(T_3G)_{10}$ and FoxP3$^{\Delta N}$ (0.2 μM) prior to streptavidin pull-down and gel analysis. Relative level of non-biotinylated DNA co-purified with biotinylated DNA was quantitated from three independent pull-downs. Two-tailed paired t-tests, in comparison to $(T_3G)_{10}$. $p < 0.001$ for ***, $p < 0.05$ for * and $p > 0.05$ for ns. c. DNA-bridging activity of FoxP3 with different combinations of $T_nG$ repeats. Biotinylated-DNA (red) and non-biotinylated DNA (blue) were mixed at 1:1 ratio and were incubated with FoxP3$^{\Delta N}$ (0.2 μM) prior to streptavidin pull-down and gel analysis of non-biotinylated DNA. $T_2G$, $T_4G$ and $T_5G$ repeats bridged better with $T_3G$ repeats than with themselves. d. DNA-binding activity of FoxP3 orthologs with indicated DNA. Experiments were performed as in Fig. 5b. e. DNA-bridging activity of FoxP3 orthologs. Experiments were performed as in Fig. 5c. f. Sequence alignment of FoxP3 orthologs from different species, showing conservation of the key interface residues (yellow highlight, arrows on top with the residue identities in *M. musculus* FoxP3). g. DNA-binding activity of FoxP3 paralogs with $T_nG$ repeats (n = 1-4). Experiments were performed as in Fig. 5b.

**Extended Data Table 1 | Table for Cryo-EM data collection, refinement and validation statistics. The statistics of the 3D reconstruction and model refinement are summarized**

## Cryo-EM data collection, refinement and validation statistics

| | #1 Decamer of FOXP3$^{\Delta N}$-TTTG18 (EMDB-40737) (PDB 8SRP) | #2 Tetramer of FOXP3$^{\Delta N}$-TTTG18 (EMDB-40736) (PDB 8SRO) |
|---|---|---|
| **Data collection and processing** | | |
| Magnification | 81000 | 81000 |
| Voltage (kV) | 300 | 300 |
| Electron exposure (e–/Å$^2$) | 60 | 60 |
| Defocus range (μm) | -0.7 to -2.1 | -0.7 to -2.1 |
| Pixel size (Å) | 0.844 | 0.844 |
| Symmetry imposed | C1 | C1 |
| Initial particle images (no.) | 4201166 | 4201166 |
| Final particle images (no.) | 317175 | 317175 |
| Map resolution (Å) | 3.6 | 3.3 |
| FSC threshold | 0.143 | 0.143 |
| | | |
| **Refinement** | | |
| Initial model used (PDB code) | 7TDX | 7TDX |
| Model resolution (Å) | 4.0 | 3.7 |
| FSC threshold | 0.5 | 0.5 |
| Map sharpening *B* factor (Å$^2$) | -168.5 | -161 |
| Model composition | | |
| Non-hydrogen atoms | 9516 | 4132 |
| Protein residues | 849 | 345 |
| Ligands | 214 | 72 |
| R.m.s. deviations | | |
| Bond lengths (Å) | 0.004 | 0.003 |
| Bond angles (°) | 0.658 | 0.559 |
| Validation | | |
| Clashscore | 9.34 | 8.41 |
| Poor rotamers (%) | 0.47 | 0.90 |
| Ramachandran plot | | |
| Favored (%) | 93.49 | 94.66 |
| Allowed (%) | 6.51 | 5.34 |
| Disallowed (%) | 0 | 0 |

# Reporting Summary

## Statistics

For all statistical analyses, confirm that the following items are present in the figure legend, table legend, main text, or Methods section.

| n/a | Confirmed | |
|---|---|---|
| ☐ | ☒ | The exact sample size (*n*) for each experimental group/condition, given as a discrete number and unit of measurement |
| ☐ | ☒ | A statement on whether measurements were taken from distinct samples or whether the same sample was measured repeatedly |
| ☐ | ☒ | The statistical test(s) used AND whether they are one- or two-sided *Only common tests should be described solely by name; describe more complex techniques in the Methods section.* |
| ☒ | ☐ | A description of all covariates tested |
| ☒ | ☐ | A description of any assumptions or corrections, such as tests of normality and adjustment for multiple comparisons |
| ☐ | ☒ | A full description of the statistical parameters including central tendency (e.g. means) or other basic estimates (e.g. regression coefficient) AND variation (e.g. standard deviation) or associated estimates of uncertainty (e.g. confidence intervals) |
| ☐ | ☒ | For null hypothesis testing, the test statistic (e.g. *F*, *t*, *r*) with confidence intervals, effect sizes, degrees of freedom and *P* value noted *Give P values as exact values whenever suitable.* |
| ☒ | ☐ | For Bayesian analysis, information on the choice of priors and Markov chain Monte Carlo settings |
| ☒ | ☐ | For hierarchical and complex designs, identification of the appropriate level for tests and full reporting of outcomes |
| ☒ | ☐ | Estimates of effect sizes (e.g. Cohen's *d*, Pearson's *r*), indicating how they were calculated |

*Our web collection on statistics for biologists contains articles on many of the points above.*

## Software and code

Policy information about availability of computer code

| Data collection | FACS data was collected by BD FACSDiva™ Software v. 6.1.3 |
|---|---|
| Data analysis | Softwares and versions: Trimmomatic (version 0.36); Bowtie2 (version 2.3.4.3); Macs2 (version 2.1.1.20160309); DiffBind (version 4.3); Bedtools (version 2.30.0); Meme Fimo (version 5.3.3); Samtools (version 1.9); Deeptools (version 3.0.2 ) |

For manuscripts utilizing custom algorithms or software that are central to the research but not yet described in published literature, software must be made available to editors and reviewers. We strongly encourage code deposition in a community repository (e.g. GitHub). See the Nature Portfolio guidelines for submitting code & software for further information.

## Data

Policy information about availability of data

All manuscripts must include a data availability statement. This statement should provide the following information, where applicable:
- Accession codes, unique identifiers, or web links for publicly available datasets
- A description of any restrictions on data availability
- For clinical datasets or third party data, please ensure that the statement adheres to our policy

Naked genomic DNA pulldown-seq, nucleosome pulldown-seq, FoxP3 mRNA-seq and FoxP3 ChIP-seq data have been deposited to the GEO database with the accession code of GSE243606. The structures and cryo-EM maps have been deposited to the PDB and the EMDB under the accession codes of 8SRP and EMD-40737

April 2023

for the decameric FoxP3 in complex with DNA, and 8SRO and EMD-40736 for the central FoxP3 tetramer in complex with DNA (focused refinement). Other research materials reported here are available on request.

# Research involving human participants, their data, or biological material

Policy information about studies with human participants or human data. See also policy information about sex, gender (identity/presentation), and sexual orientation and race, ethnicity and racism.

| Reporting on sex and gender | N/A |
|---|---|
| Reporting on race, ethnicity, or other socially relevant groupings | N/A |
| Population characteristics | N/A |
| Recruitment | N/A |
| Ethics oversight | N/A |

Note that full information on the approval of the study protocol must also be provided in the manuscript.

# Field-specific reporting

Please select the one below that is the best fit for your research. If you are not sure, read the appropriate sections before making your selection.

☒ Life sciences          ☐ Behavioural & social sciences          ☐ Ecological, evolutionary & environmental sciences

For a reference copy of the document with all sections, see nature.com/documents/nr-reporting-summary-flat.pdf

# Life sciences study design

All studies must disclose on these points even when the disclosure is negative.

| Sample size | For the FoxP3 transcriptional activity assay in CD4+ T cells, we utilized 0.2 million activated CD4+ T cells per well in a 96-well plate for each sample. For FoxP3 ChIP-seq analysis, we collected 5 million CD4+ T cells for each sample. For mRNA-seq analysis, we sorted and collected 1 million transduced CD4+ T cells, and then performed total RNA extraction for each sample.<br>The specific sample size for each experiment is described in individual figure legends. Each data point on the graphs represents an individual sample. |
|---|---|
| Data exclusions | None |
| Replication | All biochemical assays, pull-downs, bridging assays, FoxP3 transcriptional activity assay and T cell suppression assays were performed over 3 individual times. Only experimental data that were successfully replicated in all attempts are reported. See Figure legends for details. |
| Randomization | For immunofluorescence microscopy, images were taken at random locations on the cover slip. For negative-stain electron microscopy, images were also taken at random locations throughout the grid. Other experiments in this study were not subjected to randomization as the identity of the samples are predetermined during experiments; the experimental results would not be interpretable if these samples were randomized. |
| Blinding | When performing experiments, there were no samples that could be blinded as the identity of the samples are predetermined during experiments. |

# Reporting for specific materials, systems and methods

We require information from authors about some types of materials, experimental systems and methods used in many studies. Here, indicate whether each material, system or method listed is relevant to your study. If you are not sure if a list item applies to your research, read the appropriate section before selecting a response.

## Materials & experimental systems

| n/a | Involved in the study |
|:---:|:---|
| ☐ | ☒ Antibodies |
| ☐ | ☒ Eukaryotic cell lines |
| ☒ | ☐ Palaeontology and archaeology |
| ☐ | ☒ Animals and other organisms |
| ☒ | ☐ Clinical data |
| ☒ | ☐ Dual use research of concern |
| ☒ | ☐ Plants |

## Methods

| n/a | Involved in the study |
|:---:|:---|
| ☐ | ☒ ChIP-seq |
| ☐ | ☒ Flow cytometry |
| ☒ | ☐ MRI-based neuroimaging |

# Antibodies

| | |
|:---|:---|
| Antibodies used | Rabbit anti-HA C29F4 (Cell Signaling Technology, Cat#3724S), Mouse MBP Tag antibody(8G1) (Cell Signaling Technology, Cat#2396), PE anti-mouse CD4 Antibody (Biolegend, Cat#100408), APC/Cyanine7 anti-rat CD90/mouse CD90.1 (Thy-1.1) Antibody (Biolegend, Cat#202520), Brilliant Violet 421™ anti-mouse CD152 Antibody (Biolegend, Cat#106311), Pacific Blue™ anti-mouse CD25 Antibody (Biolegend, Cat#102022), Ultra-LEAF™ Purified anti-mouse CD3ε Antibody (Biolegend, Cat#100340), Ultra-LEAF™ Purified anti-mouse CD28 Antibody (Biolegend, Cat#102116), Anti-rabbit IgG HRP-linked Antibody (Cell Signaling Technology, Cat#7074) |
| Validation | All primary antibodies were validated previously by the manufacturer. Citations of studies using these antibodies and user ratings are provided on the manufacturer's websites:<br>Rabbit anti-HA C29F4 (https://www.cellsignal.com/products/primary-antibodies/ha-tag-c29f4-rabbit-mab/3724?cart-quantity=0&logged_in=0&shop-eu-id=mv6dcjq4frogokejieqbp7fcamip6pfp&N=0+4294956287&Nrpp=200&No=3000&fromPage=plp);<br>Mouse MBP Tag antibody (https://www.cellsignal.com/products/primary-antibodies/mbp-tag-8g1-mouse-mab/2396);<br>PE anti-mouse CD4 Antibody (https://www.biolegend.com/en-gb/productstab/pe-anti-mouse-cd4-antibody-250?GroupID=BLG4745);<br>APC/Cyanine7 anti-rat CD90/mouse CD90.1 (Thy-1.1) Antibody (https://www.biolegend.com/en-gb/search-results/apc-cyanine7-anti-rat-cd90-mouse-cd90-1-thy-1-1-antibody-5157?GroupID=BLG10566);<br>Brilliant Violet 421™ anti-mouse CD152 Antibody (https://www.biolegend.com/en-gb/search-results/brilliant-violet-421-anti-mouse-cd152-antibody-7322?GroupID=BLG10448);<br>Pacific Blue™ anti-mouse CD25 Antibody (https://www.biolegend.com/en-gb/products/pacific-blue-anti-mouse-cd25-antibody-3315?GroupID=BLG10428);<br>Ultra-LEAF™ Purified anti-mouse CD3ε Antibody (https://www.biolegend.com/fr-lu/cell-health/ultra-leaf-purified-anti-mouse-cd3epsilon-antibody-7722?GroupID=BLG6744);<br>Ultra-LEAF™ Purified anti-mouse CD28 Antibody (https://www.biolegend.com/fr-fr/products/ultra-leaf-purified-anti-mouse-cd28-antibody-7733?GroupID=BLG1565);<br>Anti-rabbit IgG HRP-linked Antibody (https://www.cellsignal.com/products/secondary-antibodies/anti-rabbit-igg-hrp-linked-antibody/7074). |

# Eukaryotic cell lines

Policy information about cell lines and Sex and Gender in Research

| | |
|:---|:---|
| Cell line source(s) | The A549 parental cell line was a kind gift from Dr. Susan Weiss (U Penn) and was authenticated by her lab to be the same as ATCC CCL-185. HEK293T cells were purchased from ATCC (CRL-11268). EL4 cell line was a gift from Dr.Christophe Benoist lab (Harvard Medical School) to be the same as ATCC TIB-39. |
| Authentication | No form of authentication was used for these cell lines |
| Mycoplasma contamination | These cells were verified to be mycoplasma free by using the MycoAlert Mycoplasma Detection Kit (Lonza, Cat. No. LT07-318). |
| Commonly misidentified lines (See ICLAC register) | No commonly misidentified lines were used in this study. |

# Animals and other research organisms

Policy information about studies involving animals; ARRIVE guidelines recommended for reporting animal research, and Sex and Gender in Research

| | |
|:---|:---|
| Laboratory animals | C57BL/6N mice, sourced from Taconic Biosciences, were housed in an individually ventilated cage system at the specific-pathogen-free New Research Building facility of Harvard Medical School. The mice were maintained at a controlled environment with a temperature of 20-22°C, humidity ranging from 40-55%, and a 12-hour light-dark cycle. The spleens of 12~14 weeks old female C57BL/6 mice were isolated for the study. |
| Wild animals | The study did not involve wild animals. |
| Reporting on sex | To keep consistency, 12~14 weeks old female C57BL/6 mice were used for the study. |

| Field-collected samples | The study did not involve samples collected from the field. |
| Ethics oversight | Harvard Medical Area (HMA) Standing Committee on Animals |

Note that full information on the approval of the study protocol must also be provided in the manuscript.

# Plants

| Seed stocks | N/A |
| Novel plant genotypes | N/A |
| Authentication | N/A |

# ChIP-seq

## Data deposition

☒ Confirm that both raw and final processed data have been deposited in a public database such as GEO.

☒ Confirm that you have deposited or provided access to graph files (e.g. BED files) for the called peaks.

| Data access links
*May remain private before publication.* | To review GEO accession GSE232754:
Go to https://www.ncbi.nlm.nih.gov/geo/query/acc.cgi?acc=GSE232754
Enter token qzinugqcdrunjqn into the box |
| Files in database submission | Input for naked DNA pulldown rep1
Input for naked DNA pulldown rep2
MBP-pulldown naked DNA rep1
MBP-pulldown naked DNA rep2
FoxP3-pulldown naked DNA rep1
FoxP3-pulldown naked DNA rep2

Input for nucleosome pulldown rep1
Input for nucleosome pulldown rep2
MBP-pulldown nucleosome rep1
MBP-pulldown nucleosome rep2
FoxP3-pulldown nucleosome rep1
FoxP3-pulldown nucleosome rep2

EV ChIPseq rep1
EV ChIPseq rep2
WT ChIPseq rep1
WT ChIPseq rep2
R356E ChIPseq rep1
R356E ChIPseq rep2
R396E ChIPseq rep1
R396E ChIPseq rep2
V408M ChIPseq rep1
V408M ChIPseq rep2
AAA ChIPseq rep1
AAA ChIPseq rep2 |
| Genome browser session
(e.g. UCSC) | UCSC Mouse mm10 |

## Methodology

| Replicates | 2 replicates |
| Sequencing depth | 20M PE-150 paired-end reads |
| Antibodies | Mouse MBP Tag antibody(8G1) (Cell Signaling Technology, Cat#2396);
Rabbit anti-HA C29F4 (Cell Signaling Technology, Cat# 3724S) |
| Peak calling parameters | macs2 callpeak -c control.bam -t test.bam -p 0.01 -f BAM -g mm -n test_vs_control_0.01
macs2 callpeak -c control.bam -t test.bam -p 0.05 -f BAM -g mm -n test_vs_control_0.05 |
| Data quality | First we do the quality QC test to ensure the sequencing quality using FastQC; Use Trimmomatic to remove adapters (LEADING:3 TRAILING:3 MINLEN:15); Use Bowtie2 to align reads to the genome, and the alignment rates are all over 89%. |

For FoxP3vsInput, 134,561 peaks are called at FDR 5%; For FoxP3vsMBP, 181,090 peaks are called at FDR 5%;

| Software | Trimmomatic (version 0.36); Bowtie2 (version 2.3.4.3); Macs2 (version 2.1.1.20160309); DiffBind (version 4.3); Bedtools (version 2.30.0); Meme Fimo (version 5.3.3); Samtools (version 1.9); Deeptools (version 3.0.2 ) |
|---|---|

# Flow Cytometry

## Plots

Confirm that:

☒ The axis labels state the marker and fluorochrome used (e.g. CD4-FITC).

☒ The axis scales are clearly visible. Include numbers along axes only for bottom left plot of group (a 'group' is an analysis of identical markers).

☒ All plots are contour plots with outliers or pseudocolor plots.

☒ A numerical value for number of cells or percentage (with statistics) is provided.

## Methodology

| Sample preparation | Naïve CD4+ T cells were isolated by negative selection from mouse spleens using the isolation kit (Miltenyi Biotec) according to the manufacturer's instructions. The purity was estimated to be >90% as measured by PE anti-CD4 (Biolegend) staining and FACS analysis. Naïve CD4+ T cells were then activated with anti-CD3 (Biolegend), anti-CD28 (Biolegend) and 50 U/mL of IL2 (Peprotech) in complete RPMI medium (10% FBS heat-inactivated, 2 mM L-Glutamine, 1 mM Sodium Pyruvate, 100 μM NEAA, 5 mM HEPES, 0.05 mM 2-ME). The activation state of T cells was confirmed with increased cell size and CD44 (BioLegend) expression by FACS.  After 48 hours, cells were spin-infected with retrovirus using supernatant from HEK293T cells that were transfected with retroviral plasmids including MSCV-IRES-Thy1.1 expressing FoxP3. T cells were cultured in complete RPMI medium with 100 U/mL of IL2 for 2~3 days before harvesting and analysis. |
|---|---|
| Instrument | FACSCanto; SELF-FACSARIA II |
| Software | BD FACSDiva and FlowJo |
| Cell population abundance | The purity of the isolated CD+T cells was estimated to be >90% as measured by PE anti-CD4 (Biolegend) staining and FACS. analysis. |
| Gating strategy | When analyzing CD25 or CTLA4, a homogenous population were first gated on FSC-A/SSC-A, then singlets were gated on FSC-A/FSC-H, and then cells were plotted as CD4+ vs CD4- T cells via PE staining. Gates for FoxP3-transduced CD4+T cells were determined by comparing to mock transduced samples. |

☒ Tick this box to confirm that a figure exemplifying the gating strategy is provided in the Supplementary Information.

