## [Peer Review File · Nature]

Manuscript Title: FoxP3 recognizes microsatellites and bridges DNA through multimerization

Reviewer Comments & Author Rebuttals

Reviewer Reports on the Initial Version:

Referee expertise:

Referee #1: structure, transcription factor and nucleosome binding

Referee #2: Tregs, immunology

Referee #3: transcription factor and DNA interaction

Referees' comments:

Referee #1 (Remarks to the Author):

In this manuscript analyze binding of the FoxP3 and few other TFs of Forkhead family to TnG repeats. Binding to repeats leads to Foxp3 multimerization on DNA and bridging of 2 DNAs. The works is interesting and well done, however, the importance of some of the finding remains unclear.

1. It is not clear to me what is the difference in the analysis of published ChIPseq datasets. Although I am unfamiliar with original studeis, my understanding is that the authors did not initially identify this motif. Instead, this motif was only identified in the present study. Could the authors elaborate on why this seemingly dominant motif was overlooked in previous datasets, and how they managed to recognize it now?

Can they present additional data detailing the locations and frequency of these repeats within the genome? Also, what is the length of these repeats?

If the TnG repeats in the genome are not as uniform as the ones used in the study, it would be beneficial for the authors to also test binding assays with various native repetitive DNA sequences containing different variations.

2. Considering that DNA within cells is packed into chromatin, it would be important to examine how FoxP3 interacts with chromatinized templates. It remains unclear if FoxP3 can create this type of structure in the presence of nucleosomes. If the FoxP3 multimer and DNA bridging are limited to linker DNA, it could impact the multimer's length and consequently affect bridging. The authors need to explore whether FoxP3 can bind to nucleosomal DNA or if its binding capacity is restricted to linker DNA.

3. The authors' discovery that FoxP1 and FoxP4 (and potentially other TFs of the same family) show

different motifs across various cell lines is intriguing. For instance, in lymphoma cell lines and mouse neural stem cells, they identified TnG as the most significant motif for FoxP1, whereas this motif was not found in V-Cap and K-562 cell lines. These findings suggest some form of switch in FoxP3 binding, possibly due to changes in motif accessibility in specific cell lines or changes in FoxP3 multimerization properties, which based on the authors' data, should influence binding to TnG motifs. Certain post-translational modifications could affect this multimerization. Could the authors provide further data elucidating how this switch mechanism is regulated? This would reveal regulation in binding to TnG repeats and support functional importance of their findings.

Referee #2 (Remarks to the Author):

In this manuscript, Dr. Hur and colleagues revealed a new mode for the transcription factor Foxp3 to interact with DNA. By biochemical and CryoEM approaches, the authors showed that the Foxp3 protein and TnG repeat microsatellite sequences could aggregate to form a ladder-like structure resulting in DNA bridging, with Foxp3 dimers forming the “rungs” of the ladder. Mutations disrupting the intra-rung interface did not affect Foxp3 binding to the classic forkhead consensus motif, while diminishing its binding to TnG repeat, the formation of DNA bridging, and compromising Treg cell function. Finally, the authors broadened their findings to other Foxp3 orthologs from zebrafish to humans, as well as other Foxp family members (Foxp1, 2, and 4).

Overall, this study could bring a lot of excitement to the Foxp3/Treg field as well as the TF/gene regulation field. The ladder-like architecture formed by Foxp3 and microsatellite DNA repeats is a novel model for transcription factor-DNA interaction that has not been described previously. It also could potentially answer a question lingering in the Foxp3/Treg biology field for more than a decade — when the first Foxp3 ChIP-seq was done, we were surprised that the enrichment of the forkhead consensus motif was ranked lower than quite a few other TF binding motifs within the Foxp3 binding sequences genome-wide, raising the question how important direct DNA binding contributes to Foxp3 function (Samstein RM, Cell, 2012). In retrospect, the TnG repeats did not show up in the motif analysis probably because they were classified as “junk” DNA sequences and purged by the algorithm. The bridged DNA architecture formed by the Foxp3 multimerization could also be a new model for TFs to engage in DNA looping and 3-D genome organization. The approaches used in this study were rigorous, and the data and the conclusions are convincing in general. Despite the enthusiasm this study could bring to the field, the reviewer identified several gaps in the current manuscript as listed below.

Major points:

1. From the authors' previous study (Leng F, Immunity, 2022) and the current study, Foxp3 seems to have two distinct modes to interact with DNA, the head-to-head dimers binding to the IR-FKHM motif, and the ladder-like architecture binding to TnG repeats. What is unclear is the contributions of these two operating modes to Foxp3's function in Treg cells. Since the authors generated Foxp3 mutations that are specific to each Foxp3 activity, a comparison of Tregs carrying these mutants on genome-wide Foxp3 binding and gene expression profiles should be performed. Do specific mutations in human IPEX patients correspond to Foxp3's IR-FKHM or TnG repeat binding capacity? If

so, are there any traits in IPEX patients associated with defects in either IR-FKHM or TnG repeat binding?

2. How does DNA bridging formed by the ladder-like Foxp3 multimer affect the 3-D genome structure and/or gene expression? This is the most tantalizing function the new ladder-like architecture could serve, and the authors need to elaborate more on this point. A large body of data was published on Foxp3 DNA binding, Treg gene expression, chromatin structure, and DNA looping. From mining these data, could the authors show examples that Foxp3 binding to microsatellite DNA repeat sites correlates with specific DNA looping in Tregs? Is it possible to mutate specific microsatellite sites to abolish Foxp3 binding to see if it also affects DNA looping and/or gene expression?

3. The study showed that all four P family members of the forkhead genes Foxp1 to Foxp4 share the similar feature of binding to TnG repeats and presumably form ladder-like architecture similar to Foxp3. In Treg cells, Foxp1 has been shown to bind to a large number of Foxp3 binding sites (Konopacki C, Nat Immunol, 2019). Does this mean that Foxp1 and Foxp3 play largely redundant roles in their capacity to bind to DNA satellite sites and promote DNA bridging in Treg cells?

Minor points:

1. Fig 1b and ED Fig. 1b showed de novo motif analysis of Foxp3 ChIP-seq and Cut&Run data from previous studies. However, the motif enrichment outputs appeared to be quite different between the two datasets (CnR/ChIP-seq in Fig 1b vs Sakaguchi/Rudensky ChIP-seq in ED Fig 1b). Is there an explanation for these differences? Could the authors list all the enriched motifs from the de novo analysis to show a more complete picture?

2. Please provide quantification of band intensity in the pulldown assays shown in Fig. 3c, Fig. 4d, and 4e.

Referee #4 (Remarks to the Author):

This manuscript describes the biochemical, structural, and genome-wide binding of FoxP3 to DNA. This work has made some unique and fascinating discoveries in that FoxP3 was found to assemble onto TnG microsatellite repeat DNA, that is widespread in mammalian genomes. There, FoxP3 binds as a dimer that bridge two TnG elements, plus further assembly into a pentamer of these dimers. Each dimer forms a rung between the two DNA molecules. The structure is convincing, intriguing, and is distinct from previously described head-to-head dimers. The structure provides new and valuable information on microsatellite DNA, what interacts with it, and what these structures look like. The structure also explains how the microsatellite DNA repeats need not be perfect repeats to explain FoxP3 binding. All of the biochemical, mutagenesis, and cryo-EM results and analyses are convincing, appropriately applied, and appropriately interpreted. This is a solid, high-dimensional study, that will be of broad interest and high impact.

Minor Comments

1. Abstract: I suggest adding “with each pair forming a rung” after “five pairs of FoxP3 molecules”
2. Extended Data Figs. 1c, 1d: I think I understand ED_1d, but it is not clear what the x-axis reflects in ED_1c, and why this figure is not redundant with ED_1d.
3. “Out of 196 sites showing allelic imbalance (fold change ≥ 4) in FoxP3 CNR-seq, 78 sites harbored TnG repeat-like elements in at least one allele.” What is expected by chance (p-value?)
4. “Out of the 50 pairs of sequences tested, pull-down efficiency of 47 pairs recapitulated differential binding in CNR-seq (Figs. 1c, 1d).” Is B6 being used as the reference genome for B6 mapping? And Cast reference genome being used for Cast mapping?
5. In regard to TnG microsatellite recognition being conserved among FoxP3 homologs, can the authors additionally say something about the conservation of TnG microsatellite repeats in the species examined?
6. The use of “pioneering” on page 8 may be confusing since it is a term used for other Fox proteins binding to nucleosomes. Its use in the context of FoxP3 however is different.
7. Fig 3d. Please explain “MFI” in the legend.
8. Fig. 3e. The x-axis is not legible.
9. Fig. 5d. I do not think the motif enrichment for FoxP4 is significant. The authors implication that it is significant undermines their conclusion that the ladder-like structure is particular to FoxP3 (and not other Fox proteins like FoxP4).
10. The Methods section is lacking in description of Extended Data Table 4, including an explanation of the different “v..” tabs in the Excel workbook.

Author Rebuttals to Initial Comments:

We thank all three reviewers for the thoughtful comments and suggestions. We have performed additional experiments and analyses to strengthen the manuscript. Revised texts are colored red.

Summary of key changes in Figures and Tables

Fig 1d (new): Plot of TnG repeat lengths at genomic sites with allelic bias in FoxP3 occupancy.

Fig 2f (new): Contribution of TnG repeat sequences to FoxP3-bound chromatin contacts.

Fig 3c (revised): Quantitation added.

Fig 4d (revised): Quantitation added.

ED Fig 1a, 1b (new): Genomic analysis of TnG repeat elements

ED Fig 1d, 1e, 1f (new): ChIP-seq (H3K4me3 and H3K27ac) and ATAC-seq analysis comparison of TnG repeats genome-wide vs. FoxP3-bound.

ED Fig 2c (new): FoxP3 binding to nucleosomal DNA

ED Fig 7b, 7c (new): H3K27ac ChIP-seq and ATAC-seq analysis of FoxP3-bound TnG sites vs. non-TnG sites.

ED Fig 7d, 7e (new): FoxP3-bound chromatin contact analysis.

ED Fig 8a (new): FoxP3 RNA-seq (WT vs. mutants)

ED Fig 8b (new): FoxP3 ChIP-seq (WT vs. mutants)

ED Fig 9a (new): Impact of RBR mutations on TnG repeat binding

ED Fig 9b (revised): Quantitation added and moved from Fig 4.

ED Table 1 (new): Summary of *de novo* motif analysis for FoxP3 PD-seq. New results with PD-seq using nucleosomal DNA was added. Replacing the original ED Fig. 1a.

ED Table 2 (new): Summary of *de novo* motif analysis of 2 CNR and 2 ChIP-seq data for FoxP3. Replacing the original ED Fig 1b.

ED Table 6 (new): List of FoxP3-bound chromatin contacts utilizing TnG repeats.

Referees' comments:

Referee #1 (Remarks to the Author):

In this manuscript analyze binding of the FoxP3 and few other TFs of Forkhead family to TnG repeats. Binding to repeats leads to Foxp3 multimerization on DNA and bridging of 2 DNAs. The works is interesting and well done, however, the importance of some of the finding remains unclear.

1. It is not clear to me what is the difference in the analysis of published ChIPseq datasets. Although I am unfamiliar with original studeis, my understanding is that the authors did not initially identify this motif. Instead, this motif was only identified in the present study. Could the authors elaborate on why this seemingly dominant motif was overlooked in previous datasets, and how they managed to recognize it now?

- The reviewer is correct in that we used previously available ChIP-seq and CNR-seq data published by others, but TnG repeats were overlooked in these original studies and have never been reported. We believe that this is due to the common practice of discarding highly repetitive sequences in motif analysis as they are often considered “junk” DNA or false positive. We believe that the same scenario may have been in play for other FoxP TFs, where the previous studies failed to report TnG motifs despite their strong enrichment in previously published ChIP-seq data. We now explicitly mention this point in the main text (page 3, 2nd paragraph).

Can they present additional data detailing the locations and frequency of these repeats within the genome? Also, what is the length of these repeats?

- We performed the genome-wide analysis of TnG repeats in human, mouse and zebrafish. We found that in all three species, TnG repeats are present in ~0.02-0.06% of the genome (18K for human, 46K for mouse and 5K for zebrafish that match 29 nt TnG motif using the program FIMO). The length distribution of the TnG repeat regions showed that the majority are <54 nt in length, while only a small fraction are >55 nt (in new ED Fig 1a). Genomic feature analysis showed that most T_nG-like repeats are located distal to TSSs in all three genomes of human, mouse and zebrafish (new ED Fig 1b). However, greater fractions of T_nG repeats are located within ~3 kb of TSS in higher eukaryotes (12.66%, 9.50% and 5.72% for human, mouse and zebrafish, respectively). This is despite the fact that all three species have similar genes-to-genome size ratios (see Table in ED Fig. 1a). This observation suggests that T_nG repeats may have been coopted for transcriptional functions in higher eukaryotes. These findings are now explicitly discussed on page 3 (last paragraph) and page 8 (1st paragraph).

If the TnG repeats in the genome are not as uniform as the ones used in the study, it would be beneficial for the authors to also test binding assays with various native repetitive DNA sequences containing different variations.

- In the original manuscript, we presented the results from the binding and bridging assays using 100 native repetitive DNA sequences. A few examples are shown in Fig. 1c, 1e and 4e. The complete results are summarized in ED Table 3.

2. Considering that DNA within cells is packed into chromatin, it would be important to examine how FoxP3 interacts with chromatinized templates. It remains unclear if FoxP3 can create this type of structure in the presence of nucleosomes. If the FoxP3 multimer and DNA bridging are limited to linker DNA, it could impact the multimer's length and consequently affect bridging. The authors need to explore whether FoxP3 can bind to nucleosomal DNA or if its binding capacity is restricted to linker DNA.

- Thank you for the insightful suggestion. We performed two independent experiments to address the reviewer's comment. First, we performed FoxP3 PD-seq using the nucleosomal DNA isolated from EL4 cells. This again showed that TnG repeats is one of the most significant motifs, which is now shown in new ED Table 1. Second, we performed an *in vitro* FoxP3 binding assay using reconstituted nucleosomal DNA (new ED Fig. 2c). The result showed that FoxP3 can bind TTTG repeats pre-bound by the histone octamer, but not AAAG repeats, TGTG repeats or 601 sequence, similar to its selectivity for naked DNA (in Fig. 1g, and ED Fig 2f). It is possible that FoxP3 can disrupt the nucleosome structure on TTTG repeats. Alternatively, the nucleosome on TTTG repeats may be dynamic (as indicated by the smeary pattern of the nucleosome band with TTTG repeats, in comparison to that with the 601 sequence) and FoxP3 may exploit this dynamic property to access DNA. Regardless, the result suggests that FoxP3 can recognize TTTG repeats even in the presence of nucleosomes.

3. The authors' discovery that FoxP1 and FoxP4 (and potentially other TFs of the same family) show different motifs across various cell lines is intriguing. For instance, in lymphoma cell lines and mouse neural stem cells, they identified TnG as the most significant motif for FoxP1, whereas this motif was not found in V-Cap and K-562 cell lines. These findings suggest some form of switch in FoxP3 binding, possibly due to changes in motif accessibility in specific cell lines or changes in FoxP3 multimerization properties, which based on the authors' data, should influence binding to TnG motifs. Certain post-translational modifications could affect this multimerization. Could the authors provide further data elucidating how this switch mechanism is regulated? This would reveal regulation in binding to TnG repeats and support functional importance of their findings.

- We thank the reviewer for the insightful comment. Our manuscript demonstrates that FoxP3, in the absence of any modification, is able to form multimeric structure on TnG repeats and this also occurs in the native cellular context (Tregs, which is the only cell type known for FoxP3). In contrast, FoxP1 expresses in several different cell types and its binding to TnG repeat sequences appears cell-type-dependent, suggesting that FoxP1's ability to recognize TnG repeat sequences is regulated in certain cell types. As the reviewer pointed out, this could be due to a number of different reasons—motif accessibility, post-translational modification in FoxP1 or FoxP1's interaction with suppressor proteins etc. We agree that mechanistic investigation of such context-dependence for FoxP1 would be an interesting area of future investigation. However, since the focus of the manuscript is on FoxP3, we believe, with due respect, that it is beyond the scope of the manuscript.

Referee #2 (Remarks to the Author):

In this manuscript, Dr. Hur and colleagues revealed a new mode for the transcription factor Foxp3 to interact with DNA. By biochemical and CryoEM approaches, the authors showed that the Foxp3 protein and TnG repeat microsatellite sequences could aggregate to form a ladder-like structure resulting in DNA bridging, with Foxp3 dimers forming the “rungs” of the ladder. Mutations disrupting the intra-rung interface did not affect Foxp3 binding to the classic forkhead consensus motif, while diminishing its binding to TnG repeat, the formation of DNA bridging, and compromising Treg cell function. Finally, the authors broadened their findings to other Foxp3 orthologs from zebrafish to humans, as well as other Foxp family members (Foxp1, 2, and 4).

Overall, this study could bring a lot of excitement to the Foxp3/Treg field as well as the TF/gene regulation field. The ladder-like architecture formed by Foxp3 and microsatellite DNA repeats is a novel model for transcription factor-DNA interaction that has not been described previously. It also could potentially answer a question lingering in the Foxp3/Treg biology field for more than a decade -- when the first Foxp3 ChIP-seq was done, we were surprised that the enrichment of the forkhead consensus motif was ranked lower than quite a few other TF binding motifs within the Foxp3 binding sequences genome-wide, raising the question how important direct DNA binding contributes to Foxp3 function (Samstein RM, Cell, 2012). In retrospect, the TnG repeats did not show up in the motif analysis probably because they were classified as “junk” DNA sequences and purged by the algorithm. The bridged DNA architecture formed by the Foxp3 multimerization could also be a new model for TFs to engage in DNA looping and 3-D genome organization. The approaches used in this study were rigorous, and the data and the conclusions are convincing in general. Despite the enthusiasm this study could bring to the field, the reviewer identified several gaps in the current manuscript as listed below.

Major points:

1. From the authors' previous study (Leng F, Immunity, 2022) and the current study, Foxp3 seems to have two distinct modes to interact with DNA, the head-to-head dimers binding to the IR-FKHM motif, and the ladder-like architecture binding to TnG repeats. What is unclear is the contributions of these two operating modes to Foxp3's function in Treg cells. Since the authors generated Foxp3 mutations that are specific to each Foxp3 activity, a comparison of Tregs carrying these mutants on genome-wide Foxp3 binding and gene expression profiles should be performed. Do specific mutations in human IPEX patients correspond to Foxp3's IR-FKHM or TnG repeat binding capacity? If so, are there any traits in IPEX patients associated with defects in either IR-FKHM or TnG repeat binding?

➤ We thank the reviewer for the insightful comments. We note that, unlike the ladder-like multimerization, cellular evidence for the head-to-head dimerization is currently limited based on the available FoxP3 ChIP or CNR-seq data. That is, IR-FKHM is not enriched in ChIP or CNR peaks (which we reported in Leng et al, and also summarized in the introduction of this manuscript). Furthermore, our new result presented in this manuscript now showed that mutations that we previously reported to impair the head-to-head dimerization (RBR mutations, Leng et al) also impaired the multimerization (Fig. 4b, new ED Fig. 9a), which limits use of these mutations to probe functions of the head-to-head dimerization. Similarly, an IPEX mutation (R337Q) described in Leng et al (which induces

domain-swap dimerization) also impaired both the head-to-head dimerization (Leng et al) and the ladder-like multimerization (ED Fig. 4b). We now explicitly discuss these issues on page 8.

Unlike the head-to-head dimerization, we were able to identify mutations that selectively impair the multimerization in this manuscript. This was possible because the ladder-like multimerization utilizes multiple protein surfaces—RBR, W1, H2/H4 loop and H6—whereas the head-to-head dimerization exclusively relies on RBR, which is shared with the multimerization interface. Using the four multimerization-specific mutations (R356E, V396E, V408M and 409-411AAA), we have now performed RNA-seq and ChIP-seq (new ED Fig. 8a, 8b), in addition to the T cell suppressor assay and measurement of CTLA4 and CD25 protein levels (Fig. 3d, 3e). Note that one of these mutations, V408M, is an IPEX mutation. These mutation studies revealed that the defect in the ladder-like multimerization impaired FoxP3 transcriptional functions and its target site binding. Interestingly, the mutations reduced binding to both TnG and non-TnG sites. Exact mechanism by which FoxP3 multimerization contributes to non-TnG site ChIP signal is yet unclear, but may reflect relaxed sequence specificity of FoxP3 when bridging with strong sequences (see Fig 4) or indirect interaction through partners.

2. How does DNA bridging formed by the ladder-like Foxp3 multimer affect the 3-D genome structure and/or gene expression? This is the most tantalizing function the new ladder-like architecture could serve, and the authors need to elaborate more on this point. A large body of data was published on Foxp3 DNA binding, Treg gene expression, chromatin structure, and DNA looping. From mining these data, could the authors show examples that Foxp3 binding to microsatellite DNA repeat sites correlates with specific DNA looping in Tregs? Is it possible to mutate specific microsatellite sites to abolish Foxp3 binding to see if it also affects DNA looping and/or gene expression?

- As the reviewer recommended, we analyzed the available HiC-, PLAC-, HiChIP-, CNR-, ChIP-, ATAC-seq data. The limited resolution of HiC-, PLAC-, HiChIP-seq data (5-10kb) precluded direct motif analysis of the chromatin contact anchors. Instead, we asked how frequently contacts are made between anchors containing FoxP3 CNR peaks with T_nG repeats (T_nG anchors) vs. those lacking T_nG repeats (non-T_nG anchors). Among the high-frequency contacts between FoxP3-bound anchors, we found that those between two T_nG anchors (30-53%) were more prevalent than expected by chance (13.7%, $p < 0.0001$) and that such T_nG–T_nG contacts were more enriched among the stronger contacts (new Fig. 2f, new ED Table 6). In contrast, non-T_nG–non-T_nG contacts were more depleted among the stronger contacts. This is despite the fact that non-T_nG CNR peaks have higher levels of chromatin accessibility and H3K4me3 than T_nG CNR peaks, while displaying similar H3K27ac levels (new ED Fig. 7a-7c). Most of the T_nG–T_nG contacts showed increased frequency in WT Treg relative to in FoxP3 knock-out (new ED Fig. 7a). Furthermore, many of the anchors for the T_nG–T_nG contacts were nearby Treg signature genes (e.g. *Il2ra*, *Cd28*, *Tnfrsf3*, *Ets1* in new ED Table 6, tab 7), and were also observed in previously characterized enhancer-promoter loops in Tregs (new ED Fig. 7b), implicating their transcriptional functions. These results together support that FoxP3 multimerization on T_nG repeats contributes to long-distance chromatin contacts in Tregs.

3. The study showed that all four P family members of the forkhead genes Foxp1 to Foxp4 share the similar feature of binding to TnG repeats and presumably form ladder-like architecture

similar to Foxp3. In Treg cells, Foxp1 has been shown to bind to a large number of Foxp3 binding sites (Konopacki C, Nat Immunol, 2019). Does this mean that Foxp1 and Foxp3 play largely redundant roles in their capacity to bind to DNA satellite sites and promote DNA bridging in Treg cells?

- We again thank the reviewer for this interesting question. As the reviewer pointed out, the TnG binding and bridging activity could be redundantly carried out by FoxP3 and FoxP1. But it is also possible that these activities alone may not be sufficient to confer transcriptional effect and may need to be coupled with the cofactors that FoxP3 and FoxP1 recruit. FoxP3 is known to recruit a large number of cofactors through the N-terminal pro-rich region, which differs from the FoxP1 N-terminal region. This difference in the cofactor recruitment may diverge their functions, which may explain why FoxP1 cannot functionally substitute FoxP3 in Tregs.

Minor points:

1. Fig 1b and ED Fig. 1b showed de novo motif analysis of Foxp3 ChIP-seq and Cut&Run data from previous studies. However, the motif enrichment outputs appeared to be quite different between the two datasets (CnR/ChIP-seq in Fig 1b vs Sakaguchi/Rudensky ChIP-seq in ED Fig 1b). Is there an explanation for these differences? Could the authors list all the enriched motifs from the de novo analysis to show a more complete picture?

- The reviewer is correct that different motifs are seen depending on the data set, with TnG repeat motif one of the most consistent and most significant motifs seen across all data sets. To provide a more complete picture, we performed motif analysis (MEME-ChIP and STREME) using all four data sets available (two CnR-seq from the Rudensky and Dixon labs, and two ChIP-seq from Rudensky and Sakaguchi labs) and listed top 4 motifs for each data set and each method of analysis in new ED Table 2.

2. Please provide quantification of band intensity in the pulldown assays shown in Fig. 3c, Fig. 4d, and 4e.

- The quantitative analyses of the triplicate experiments are now shown in Figs. 3c, 4d and ED Fig 9b (original Fig 4e).

Referee #4 (Remarks to the Author):

This manuscript describes the biochemical, structural, and genome-wide binding of FoxP3 to DNA. This work has made some unique and fascinating discoveries in that FoxP3 was found to assemble onto TnG microsatellite repeat DNA, that is widespread in mammalian genomes. There, FoxP3 binds as a dimer that bridge two TnG elements, plus further assembly into a pentamer of these dimers. Each dimer forms a rung between the two DNA molecules. The structure is convincing, intriguing, and is distinct from previously described head-to-head dimers. The structure provides new and valuable information on microsatellite DNA, what interacts with it, and what these structures look like. The structure also explains how the microsatellite DNA repeats need not be perfect repeats to explain FoxP3 binding. All of the biochemical, mutagenesis, and cryo-EM results and analyses are convincing, appropriately applied, and appropriately interpreted. This is a solid, high-dimensional study, that will be of broad interest and high impact.

➤ We thank the reviewer for the positive assessment of the manuscript.

Minor Comments

1. Abstract: I suggest adding “with each pair forming a rung” after “five pairs of FoxP3 molecules”

➤ We made the change as suggested.

2. Extended Data Figs. 1c, 1d: I think I understand ED_1d, but it is not clear what the x-axis reflects in ED_1c, and why this figure is not redundant with ED_1d.

➤ We agreed with the reviewer and removed ED Fig. 1c.

3. “Out of 196 sites showing allelic imbalance (fold change ≥ 4) in FoxP3 CNR-seq, 78 sites harbored TnG repeat-like elements in at least one allele.” What is expected by chance (p-value?)

➤ The frequency of TnG repeats in the mouse genome is $\sim 0.06\%$ (ED Fig. 1a), which is significantly lower than the frequency of TnG repeats seen in allele-biased sites (39.3% , $p < 1e-8$). Furthermore, the allelic imbalance in TnG repeat length largely segregates with the allelic imbalance in FoxP3 occupancy (new Fig 1d), which makes our conclusion stronger. These points are explicitly discussed in the 3rd paragraph of page 3.

4. “Out of the 50 pairs of sequences tested, pull-down efficiency of 47 pairs recapitulated differential binding in CNR-seq (Figs. 1c, 1d).” Is B6 being used as the reference genome for B6 mapping? And Cast reference genome being used for Cast mapping?

- Both are correct. We note that the mapping was done in the original paper (van der Veecken et al, Immunity, 2020) and our analysis was based on the mapped reads.

5. In regard to TnG microsatellite recognition being conserved among FoxP3 homologs, can the authors additionally say something about the conservation of TnG microsatellite repeats in the species examined?

- We performed a genome-wide analysis of TnG repeats in human, mouse and zebrafish. We found that there are variable number of TnG repeats in the genome (18K for human, 46K for mouse and 5K for zebrafish, ED Fig. 1a). While T_nG-like repeats are located more frequently in TSS-distal regions in all three species, greater fractions of T_nG repeats are located within ~3 kb of TSS in higher eukaryotes (12.66%, 9.50% and 5.72% for human, mouse and zebrafish, respectively), even though all three species have similar genes-to-genome size ratios. This observation suggests that T_nG repeats may have been coopted for transcriptional functions in higher eukaryotes. These points are now explicitly discussed on page 8 (1st paragraph).

6. The use of “pioneering” on page 8 may be confusing since it is a term used for other Fox proteins binding to nucleosomes. Its use in the context of FoxP3 however is different.

- It is now rephrased to read “the first set of FoxP3 molecules”, rather than “pioneering FoxP3 molecules”.

7. Fig 3d. Please explain “MFI” in the legend.

- We now explain MFI (mean fluorescence intensity) in the legend.

8. Fig. 3e. The x-axis is not legible.

- We increased the font size to make the labels legible.

9. Fig. 5d. I do not think the motif enrichment for FoxP4 is significant. The authors implication that it is significant undermines their conclusion that the ladder-like structure is particular to FoxP3 (and not other Fox proteins like FoxP4).

- We agree that TnG repeat-like motif enrichment is not as strong for FoxP4 than for FoxP3. This is now explicitly stated in the main text.

10. The Methods section is lacking in description of Extended Data Table 4, including an explanation of the different “v..” tabs in the Excel workbook.

- We apologize for the oversight of including multiple unnecessary tabs. These tabs are now removed from ED Table 7 (original ED Table 4). The method for ED Table 8 is described under the heading “Motif analysis of other Forkhead TFs”.

Reviewer Reports on the First Revision:

Referees' comments:

Referee #1 (Remarks to the Author):

The authors have resolved all my issues. I have no additional concerns.

Referee #2 (Remarks to the Author):

In the revised manuscript, the authors provided a substantial amount of new data and analysis so that their conclusions are now standing on a more concrete foundation. The new Foxp3 mutant ChIP-seq data also suggested that the ladder-like structure could play a role in Foxp3 binding to the non-TnG repeat sequences, implicating an even more critical function for this Foxp3 oligomer structure. The concerns raised by this reviewer were satisfactorily addressed in the revision. I congratulate the authors on their beautiful work and look forward to the follow-up studies inspired by this study in the coming years.

Referee #4 (Remarks to the Author):

The revised manuscript is acceptable. The authors have done an excellent job on the biochemistry, genomic analysis, and atomic structure of FoxP3. The structure of the assembled complex will be of wide interest.